# Continuous, Subject-Specific Attribute Control in T2I Models by Identifying Semantic Directions

## Abstract

Recent advances in text-to-image (T2I) diffusion models have significantly improved the quality of generated images. However, providing efficient control over individual subjects, particularly the attributes characterizing them, remains a key challenge. While existing methods have introduced mechanisms to modulate attribute expression, they typically provide either detailed, object-specific localization of such a modification or fine-grained, nuanced control of attributes. No current approach offers both simultaneously, resulting in a gap when trying to achieve precise continuous and subject-specific attribute modulation in image generation. In this work, we demonstrate that token-level directions exist within commonly used CLIP text embeddings that enable fine-grained, subject-specific control of high-level attributes in T2I models. We introduce two methods to identify these directions: a simple, optimization-free technique and a learning-based approach that utilizes the T2I model to characterize semantic concepts more specifically. Our methods allow the augmentation of the prompt text input, enabling fine-grained control over multiple attributes of individual subjects simultaneously, without requiring any modifications to the diffusion model itself. This approach offers a unified solution that fills the gap between global and localized control, providing competitive flexibility and precision in text-guided image generation.

## 1 Introduction

Text-to-image (T2I) diffusion models have rapidly advanced, achieving remarkable quality in generating visually stunning images (Rombach et al., 2022; Imagen-Team, 2024). However, as the quality of generated images improves, the need for precise control over the generation process becomes increasingly crucial. This control should extend beyond simply adjusting *what* is depicted in the scene. It must also provide nuanced control of the attributes describing *how* these objects are characterized. Attributes, such as a person's age, are not binary or static – they often span a continuum, requiring models to capture fine-grained variations to produce results that align with user intent.

Currently, a fundamental gap exists: no method provides fine-grained modulation and subject-specific localization simultaneously. Recent works like Prompt-to-Prompt (P2P) (Hertz et al., 2023) and Concept Sliders (Gandikota et al., 2024) have made significant strides in introducing control into T2I models. P2P enables localized expression changes, allowing adjustments to specific aspects of a given image based on text modifications, while Concept Sliders facilitate fine-grained modulation over global attributes across all subject instances. This limitation means that while we can tweak attributes globally or localize changes to subjects, we still lack a unified, generalized approach capable of concurrently achieving fine-grained control for both aspects.

This work aims to bridge this gap by introducing a method that enables unified, subject-specific, fine-grained control over attributes within T2I diffusion models. Unlike existing methods that provide either localized coarse control or global fine-grained control, our approach offers precise modulation of attributes that can be directed at specific subjects within the generated image (see Figure 1). This results in an unprecedented level of intuitive control, allowing users to fine-tune not just what appears in an image but how it appears, down to the smallest level of attribute expression.

Figure 1: (a) Our method augments the prompt input of image generation models with *fine-grained control* of attribute expression in generated images (unmodified images are marked in **green**) in a *subject-specific* manner *without additional cost* during generation. (b, c) Previous methods only allow *either* fine-grained expression control or fine-grained localization when starting from the image generated from a basic prompt.

We summarize our main contributions as follows:

- We show that token-level edit directions exist within common CLIP embeddings, enabling fine-grained control of subject-specific attributes, and show that diffusion models can effectively interpret these directions.

- We introduce a simple, optimization-free approach to identify attribute-specific directions by contrasting text prompts that describe the desired attributes or concepts.

- We introduce a second, learning-based method that identifies more robust directions through backpropagation of high-level semantic concepts to the text embedding input, using a reconstruction loss objective.

- We show that these token-level edit directions enable fine-grained, subject-specific, compositional control of attributes and concepts in generated images.

## 2 RELATED WORK

The rapid advancements in generative models for image and video synthesis, particularly diffusion models like Stable Diffusion (Rombach et al., 2022), have spurred efforts to develop techniques for fine-grained editing and control of specific attributes in generated content. Our work focuses on enabling precise, subject-specific control in images by targeting individual characteristics in a controlled and continuous manner.

Existing methods for controlled generation and image editing can be broadly categorized based on the underlying generative models – primarily Generative Adversarial Networks (GANs) (Goodfellow et al., 2014) and Diffusion Models (Ho et al., 2020) –, and the mechanisms they use for control – typically latent space manipulations or textual descriptions.

**T2I Diffusion Model Preliminaries** T2I Diffusion models (Rombach et al., 2022; Podell et al., 2024) simulate a reverse diffusion process $p_\theta(\mathbf{x}_{0:T}|P)$ that enables sampling from the distribution of images $p_\theta(\mathbf{x}_0|P)$ given a text conditioning $P$ and a Gaussian noise sample $\mathbf{x}_T$. They iteratively denoise $\mathbf{x}_T$ using a diffusion model $\hat{\epsilon}_\theta(\mathbf{x}_t|P, t)$. This is typically done by learning to predict the noise content $\epsilon$ in the sample $\mathbf{x}_t = \alpha_t \mathbf{x}_0 + \sigma_t \epsilon$ using the following loss function:

$$\mathcal{L}_{\text{Diffusion}} = \mathbb{E}_{(\mathbf{x}_0, \mathbf{c}) \sim p_{\text{data}}(\mathbf{x}_0, \mathbf{c}), \epsilon \sim \mathcal{N}(0, \mathbf{I}), t \sim \mathcal{U}(0, T]} \left[ w(t) \left\| \epsilon - \hat{\epsilon}_\theta(\alpha_t \mathbf{x}_0 + \sigma_t \epsilon | \mathbf{c}, t) \right\|_2^2 \right], \quad (1)$$

where $\hat{\epsilon}_\theta(\cdot)$ is the diffusion model conditioned on the timestep $t$ and the conditioning signal $\mathbf{c}$, $w(t)$ is a loss weighting term, and $\alpha_t$ and $\sigma_t$ are noise schedule parameters. The conditioning $\mathbf{c}$ is typically obtained using a CLIP (Radford et al., 2021) text encoder $\mathcal{E}_{\text{CLIP}}$ as a tokenwise embedding $\mathbf{e} = \mathcal{E}_{\text{CLIP}}(P)$ of a text prompt $P$.

## 2.1 GAN-based Image Editing and CLIP-Based directions

GANs (Goodfellow et al., 2014; Radford et al., 2016), particularly StyleGANs (Karras et al., 2019), are popular for image editing due to their generative power and disentangled latent space. Methods like InterFaceGAN (Shen et al., 2020) manipulate attributes by identifying latent space directions. Approaches such as StyleCLIP (Patashnik et al., 2021), CLIP2StyleGAN (Abdal et al., 2022), and TediGAN (Xia et al., 2021) use CLIP (Radford et al., 2021) for text-based guidance in latent space editing. Despite these advancements, these methods inherit the limitations of StyleGAN and struggle to generalize to complex, multi-subject images.

## 2.2 Steering the Generation Process of Diffusion Models

**Direction-based Control**   Similar to GAN-based editing, approaches like DiffusionCLIP (Kim et al., 2022) use CLIP for editing with unconditional closed-domain diffusion models. Recent methods, such as Asyrp (Kwon et al., 2023), InterpretDiffusion (Li et al., 2024a), LFM (Hu et al., 2024), and BoundaryDiffusion (Zhu et al., 2023), modulate learned directions in the diffusion backbone or noise space, similar to StyleGAN. Concept Sliders (Gandikota et al., 2024) achieve disentangled attribute modulation by training attribute-specific LoRAs (Hu et al., 2022), however these methods typically lack subject specificity, as they perform global modulations . Mask-based approaches like MAG (Mao et al., 2023) allow more targeted control but require significant user input to define the masks.

**Attention Map-based Control**   Building on the observation by Hertz et al. (2023) that fixing attention maps during generation while changing the text prompt enables generating variations of images, a range of control methods utilizing this mechanism have been introduced. Methods like Prompt-to-Prompt (Hertz et al., 2023), MasaCtrl (Cao et al., 2023), AdapEdit (Ma et al., 2024), and many others (Brooks et al., 2023; Simsar et al., 2023; Zhang et al., 2024) leverage attention control combined with prompt editing to allow for subject-specific manipulations via text interfaces. These methods provide intuitive control and subject-specificity but suffer from the inherent discreteness of text inputs and struggle with fine-grained control over the magnitude of changes.

**From Controlled Generation to Editing**   For editing real images, inversion techniques are employed to map images back into a model's latent space. In GAN-based methods, Image2StyleGAN (Abdal et al., 2019) and In-Domain GAN Inversion (Zhu et al., 2020) are commonly used. Similarly, for diffusion models, DDIM Inversion (Dhariwal & Nichol, 2021), Null-Text Inversion (Mokady et al., 2023), and ReNoise (Garibi et al., 2024) enable mapping images to the latent noise space, allowing editing of real images via re-generation with controlled generation methods.

## 3 Method

Let $M$ denote the number of attributes we consider in our work and let $N$ denote the number of subjects mentioned in a prompt $P$, $\mathcal{A} = \{A_i \mid i \in [\![1, M]\!]\}$ denote the set of attributes $A_i$ and $\mathcal{S}_P = \{S_j \mid j \in [\![1, N]\!]\}$ denote the set of subjects $S_j$ mentioned in the prompt. We aim to influence the generation process to enable control over the expression $\mathrm{expr}(A_i)$ of specific attributes $A_i \in \mathcal{A}$ of specific subjects $S_j \in \mathcal{S}_P$. As an example, consider the prompt "a portrait of a man and woman sharing a laugh". If the man should be younger, one can change "man" to "young man", but this does not offer continuous control over how young the man is supposed to be. Instead, we aim to provide the same subject-specificity that changing the prompt offers, but without the limitations of the non-continuousness of language. Unlike previous works, we wish to provide control that is simultaneously i) continuous, ii) subject-specific, and iii) does not require manual image masks or reference images.

Our key observation is that the diffusion model's *interpretation* of the tokenwise CLIP text embedding vector $\mathbf{e} = \mathcal{E}_{\mathrm{CLIP}}(P) = (\mathbf{e}_{<\mathrm{SOS}>}, \mathbf{e}_1, \ldots, \mathbf{e}_k, \mathbf{e}_{<\mathrm{EOS}>})$, which is typically used to condition the model, is *locally* smooth and enables *subject-specific* semantic modulations (Section 3.1). Using this property, we can continuously modulate semantic attributes of specific subject instances in the prompt $P$. To enable targeted modulation of specific attributes, we introduce methods to identify latent space directions corresponding to attributes $\mathcal{A}$ (e.g., "old", "happy", "expensive").

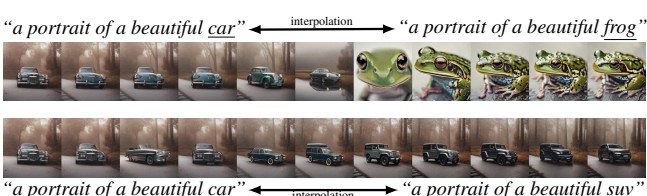

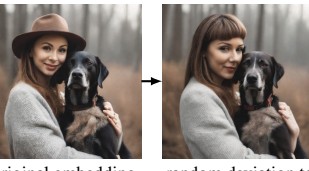

*"a portrait of a beautiful car"* ◄────── interpolation ──────► *"a portrait of a beautiful frog"*

*"a portrait of a beautiful car"* ◄────── interpolation ──────► *"a portrait of a beautiful suv"*

*"a portrait of a beautiful woman with her beautiful dog"*

original embedding    random deviation to "woman" token emb.

Figure 2: The tokenwise CLIP text embedding space is not globally smooth. We linearly interpolate between the embeddings of two prompts while keeping the noise seed fixed. Near the original embeddings, changes are smooth and semantically interpretable, but strong phase transitions exist between substantially different subjects (e.g., "car" vs. "frog").

Figure 3: The tokenwise CLIP embedding space enables subject-specific interventions. Changes to the embedding of subject tokens can lead to disentangled local changes focused on that subject.

### 3.1 INTERPRETATION OF TOKENWISE CLIP TEXT EMBEDDINGS IN DIFFUSION MODELS

**Global *v.s.* Local Behavior** Unlike the *pooled* text embedding space of CLIP (Radford et al., 2021) models, which has been explored extensively in previous works (Patashnik et al., 2021; Wang et al., 2023; Ramesh et al., 2022), the *tokenwise* text embedding space has not been investigated as much. Previous methods (Chefer et al., 2024; Li et al., 2024b; Wang et al., 2024) typically interpret this space *globally*, applying projections onto subspaces to decompose concepts or eliminate them from the generated images. Conversely, we find two distinct *local* behaviors in the tokenwise CLIP embedding space as interpreted by diffusion models (Podell et al., 2024). We can observe strong local phase changes when interpolating between substantially different subjects (see Figure 2, top row). Here minor changes in the embedding cause drastic changes in the generated images. At the same time, the space shows smooth, semantically interpretable changes in the vicinity of the original embeddings and when interpolating between similar subjects (see Figure 2, bottom row).

**Subject-Specificity** The CLIP tokenizer typically maps individual words to single tokens. Diffusion models also directly attend to adjectives added to subjects in the prompt to determine details of the subjects' appearance (Hertz et al., 2023; Rassin et al., 2023). Despite this direct connection, additional information is also stored in other tokens, especially the following tokens describing the subject, and is interpreted by the diffusion model (Li et al., 2024b). Our key observation here is that we can exploit this semantic aggregation in the subject tokens to perform targeted interventions: modulating the token embedding $\mathbf{e}_{[S_j]}$ of a specific subject $S_j$ primarily affects only that subject in the generated image (see Figure 3), without the need for adding new tokens.

### 3.2 IDENTIFYING SEMANTIC DIRECTIONS FROM CONTRASTIVE PROMPTS

To use the key observations in Section 3.1 for subject-specific control, we have to identify which directions enable modulating specific attributes. We previously found that interpolation of the tokenwise text embeddings leads to locally smooth changes around the original embeddings (c.f. Figure 2). Motivated by this finding, we propose identifying semantic directions in the tokenwise embedding space by comparing embeddings of contrastive prompts.

Formally, given a target attribute $A_i$, defined via an adjective (e.g., *"old"*), we want to identify a direction vector $\Delta\mathbf{e}_{A_i} \in \mathbb{R}^{d_{\mathrm{CLIP}}}$ that can be added to the embedding of a target subject token $\mathbf{e}_{[S_j]}$ to modulate the expression of that attribute $\expr_{S_j}(A_i)$ in the generated image. To identify this direction, we first obtain the tokenwise CLIP embeddings for two prompts: a neutral prompt $P$ describing a single subject $S$ and a positive prompt $P_+$, which prepends the adjective to the subject. Then, we compute the difference between the subject token embeddings $\mathbf{e}_{[S]}$ :

$$\Delta\mathbf{e}_{A_i} = (\mathcal{E}_{\mathrm{CLIP}}(P_+) - \mathcal{E}_{\mathrm{CLIP}}(P))_{[S]}. \tag{2}$$

This directly yields a direction $\Delta\mathbf{e}_{A_i}$ that captures the change induced by prepending the adjective to the subject noun in the text prompt. To obtain more robust estimates of this direction, we average it over a multitude of prompt pairs which describe the same target attribute $A_i$.

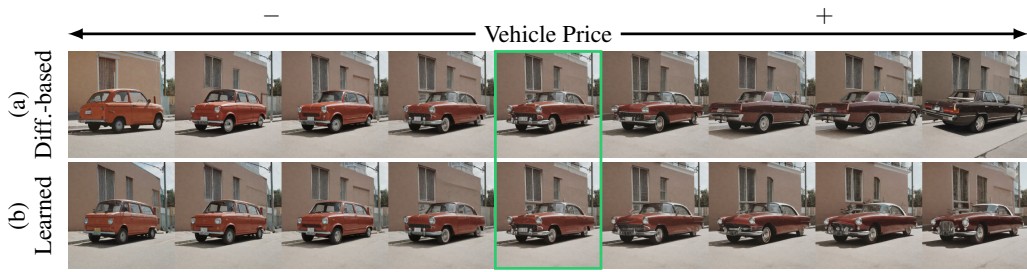

Figure 4: Variations along "vehicle price" directions identified using our methods. (a) Modulate along direction from difference-based approach (Section 3.2). (b) Modulate along direction from robust learned approach (Section 3.3). Unmodified images are marked in **green**. These directions successfully capture the target attribute and allow for fine-grained modulation but (a) also shows unwanted side-effects such as flipping the car's orientation.

To modulate that attribute's expression $\text{expr}_{S_j}(A_i)$ in the generated image for a given prompt embedding $\mathbf{e}$ and target subject $S_j$, we apply the modulation $\lambda_i \Delta\mathbf{e}_{A_i}$ to $\mathbf{e}$ with

$$\mathbf{e}'(\mathbf{e}, \lambda_i \Delta\mathbf{e}_{A_i})_{[S_j]} = \mathbf{e}_{[S_j]} + \lambda_i \Delta\mathbf{e}_{A_i}, \tag{3}$$

where $\lambda_i$ is a scalar controlling the magnitude of the modulation. This modified embedding is then passed to the diffusion model in place of $\mathbf{e}$. This omits any changes to tokens other than the target subject noun, including the `<EOS>` token, which plays a crucial role in the image generation process (Yesiltepe et al., 2024; Li et al., 2024b; Wu et al., 2024). Despite this, it successfully enables the modulation of target attributes (see Figure 4a).

### 3.3 IDENTIFYING ROBUST SEMANTIC DIRECTIONS VIA DIFFUSION NOISE PREDICTIONS

Although the simple difference-based method introduced in Section 3.2 is effective in many scenarios, it has several limitations. In practice, it often leads to unintended side effects (see Figure 4) and is limited to attributes $A_i$ expressible as prefixes to the subject noun, due to the causality of the CLIP text encoder. To address these issues, we propose a substantially more robust approach for identifying such directions. To obtain more robust directions, we use a T2I diffusion model to identify associations of adjectives to directions in the tokenwise embedding space. This effectively inverts the typical relation, where language models are used to augment the T2I model, such as with prompt augmentation (Betker et al., 2023). We use the diffusion model to identify sample-specific directions corresponding to modulations of the target attribute in the noise prediction space and backpropagate them through the diffusion model to discover *generalizable*, *fine-grained* local modulation directions $\Delta\mathbf{e}_{A_i}$ within the tokenwise CLIP embedding space. Specifically, we aim to apply the modulation and change the image similarly to adding an adjective to the prompt, but without adding additional tokens or affecting the rest of the embedding, and while enabling fine-grained modulations.

We start with a random (generated) image $\mathbf{x}_0$ and its corresponding neutral prompt $P$ describing one subject $S$ and sample a random timestep $t \sim \mathcal{U}[0, T]$. We obtain the noised latent as $\mathbf{x}_t = \alpha_t \mathbf{x}_0 + \sigma_t \boldsymbol{\epsilon}, \boldsymbol{\epsilon} \sim \mathcal{N}(0, \mathbf{I})$, where $\alpha_t$ and $\sigma_t$ are time-dependent noise schedule coefficients. Then, we predict the noise for two different prompts with the T2I diffusion model: the original prompt, $\tilde{\boldsymbol{\epsilon}} = \hat{\boldsymbol{\epsilon}}_\theta(\mathbf{x}_t | P)$ and the prompt with the adjective added, $\tilde{\boldsymbol{\epsilon}}_+ = \hat{\boldsymbol{\epsilon}}_\theta(\mathbf{x}_t | P_+)$. Using these two noise predictions, we obtain a direction $\Delta\tilde{\boldsymbol{\epsilon}} = \tilde{\boldsymbol{\epsilon}}_+ - \tilde{\boldsymbol{\epsilon}}$ in that particular image's and prompt's noise space corresponding to modulating $A_i$.[1] Finally, we distill that direction in the noise space through the diffusion model into the direction $\Delta\mathbf{e}_{A_i}$ (see Figure 5 for an illustration) using the reconstruction loss

$$\mathcal{L}(\mathbf{x}_0, \mathbf{e}; \Delta\mathbf{e}_{A_i}) = \mathbb{E}_{\lambda_i, \boldsymbol{\epsilon} \sim \mathcal{N}(0, \mathbf{I}), t \sim \mathcal{U}[0, T]} \left[ w(t) \left\| (\boldsymbol{\epsilon} + \lambda_i \Delta\tilde{\boldsymbol{\epsilon}}) - \hat{\boldsymbol{\epsilon}}_\theta(\mathbf{x}_t | \mathbf{e}'(\mathbf{e}, \lambda_i \Delta\mathbf{e}_{A_i}), t) \right\|_2^2 \right], \tag{4}$$

adapted from Equation (1). To capture the full scale of potential changes, including fine-grained ones, we randomly vary $\lambda_i$. Finally, to obtain a robust, generalizable direction for $A_i$, we optimize

---

[1]If an attribute can be described using a contrastive pair of adjectives (e.g., *"old"* and *"young"*), we use the direction $\Delta\tilde{\boldsymbol{\epsilon}} = \tilde{\boldsymbol{\epsilon}}_+ - \tilde{\boldsymbol{\epsilon}}_-$ between the noise predictions instead to increase robustness.

$\hat{\epsilon}_\theta(\cdot | \mathcal{E}(\text{happy person}))$

$\mathbf{x}_t$

$\hat{\epsilon}_\theta(\cdot | \mathcal{E}(\text{person}))$

Noise Prediction Space
(Image Space)

$\Delta \mathbf{e}_{A_i}$

$\mathcal{E}(\text{person})$

$\mathbf{x}_t \rightarrow \hat{\epsilon}_\theta(\mathbf{x}_t | \cdot) \xrightarrow{\tilde{\epsilon}} \mathcal{L}$ (Equation (4))

$\mathbf{e}_{[S_j]}$

Figure 5: Illustration of the intuition of our method. We find that directions that correspond to modulating an attribute $A_i$ in the noise prediction space $\Delta\tilde{\epsilon}$ (**green**) from a specific starting point $\mathbf{x}_t$ can be backpropagated (**purple**) through the diffusion model (Equation (4)) to obtain a generalized corresponding direction $\Delta\mathbf{e}_{A_i}$ (**blue**) in the tokenwise embedding space. $\mathcal{E}(P)$ is the prompt embedding, and $\hat{\epsilon}_\theta(\cdot)$ denotes the diffusion model.

$\Delta\mathbf{e}_{A_i}$ using AdamW (Loshchilov & Hutter, 2019) over a wide range of different sampled images $\mathbf{x}_0$ from different base prompts $P$, noises $\epsilon$, and timesteps $t$. Unlike Gandikota et al. (2024), we predict a continuous target direction and train on that by continuously varying $\lambda_i$. We provide an overview of the full training algorithm in Algorithm 1.

### 3.4 ATTRIBUTE CONTROL

During inference time, we use Equation (3) to control the expression $\text{expr}_{S_j}(A_i)$ of an attribute $A_i$ of a specific subject $S_j$. By adding the modulation $\Delta\mathbf{e}_{A_i}$ to the target subject $S_j$ in the tokenwise prompt embedding $\mathbf{e}$, we bias the distribution of generated images $p(\mathbf{x}_0)$ towards increased or decreased expression of the target attribute $A_i$ for the target subject $S_j$ (see Figure 6). We typically apply the modulation after the first 20% of sampling steps to achieve more fine-grained changes, as in (Meng et al., 2022; Gandikota et al., 2024). Moreover, this approach supports the additivity of attribute modulations, allowing for multiple simultaneous edits. By adding several modulation vectors $\Delta\mathbf{e}_{A_i}$, we can independently adjust different attributes for the same subject $S_j$ without interfering with each other. Our method also allows for editing multiple subjects within the same image by applying separate modulations to different subjects. As applying our method only requires one addition, it effectively adds zero inference cost.

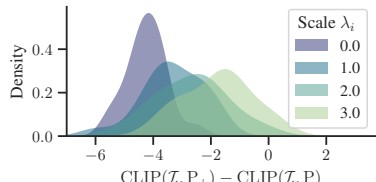

Figure 6: Applying modulations $\lambda_i\Delta\mathbf{e}_{A_i}$ gradually shifts the distribution of generated images w.r.t. the expression of the target attribute $\text{expr}(A_i)$. We show the kernel density estimation of the CLIP score difference between *"a photo of an expensive car"* & *"a photo of a car"* (original prompt) while modulating $\text{expr}_{\text{car}}(\text{vehicle price})$.

**Application to Real Image Editing**     In addition to modulating attributes in generated images, our method can also be used to perform fine-grained edits of real images. We first invert the given real image $\mathcal{I}$ with a matching caption (obtained, e.g., by user input or synthetic captioning) into its corresponding noise latent $\mathbf{x}_T$ using an off-the-shelf inversion method (Garibi et al., 2024). Then, we regenerate the image while applying our attribute modulation to the target subject in the same manner as when generating images from scratch to obtain fine-grained subject-specific edits of real images.

## 4 EXPERIMENTS

In this section, we comprehensively evaluate our proposed method. We conduct experiments by applying our semantic directions to both biasing the distribution of generated images and editing real images. We validate key properties such as subject specificity, the disentanglement of edits, the fine-grainedness of control, and inference performance.

### 4.1 EXPERIMENTAL SETUP

We evaluate our proposed method primarily on Stable Diffusion XL (Podell et al., 2024), a widely used large-scale T2I diffusion model. To test our method, we obtain a large variety of semantic direc-

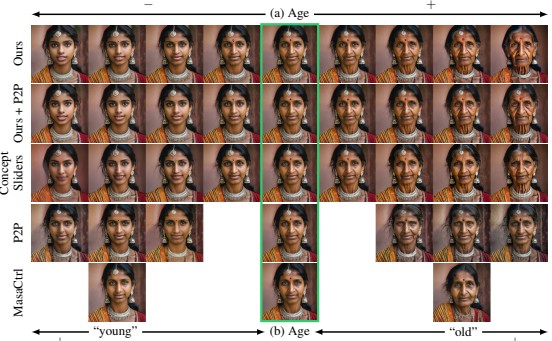

Figure 7: Our modulations allow fine-grained control of many attributes over many categories. Unmodulated images are marked in **green**. As the changes are fine-grained and smooth, we recommend zooming in.

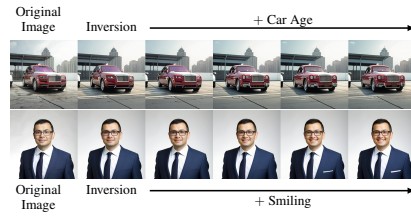

Figure 9: Real image editing: we apply our method to editing by inverting the image with ReNoise (Garibi et al., 2024) and regenerating the image with our modulations applied.

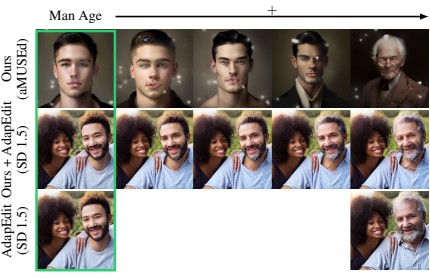

Figure 8: Qualitative comparison with other methods. (a) We continuously modulate the age of a person. (b) P2P (Hertz et al., 2023) and MasaCtrl (Cao et al., 2023) do not offer full continuous control, first modulating to "old" or "young" and then optionally reweighting the adjective from there in the case of P2P. Unmodulated images are marked in **green**.

Figure 10: Zero-shot transfer: our modulations can be learned on one model (SDXL) and transferred to others (including non-diffusion models) without retraining. This also allows us to combine them with methods for other models, such as AdapEdit (Ma et al., 2024) on SD 1.5, which does not offer continuous subject-specific modulations by itself. Unmodulated images are marked in **green**.

tions for various attributes, primarily focused on humans, but also including vehicles and furniture. Detailed training procedures and parameters are in Appendix B.1.

**Integration with other methods** As our modulations augment the text prompt embedding input without adapting the model, they can directly be combined with many controlled generation and editing methods that utilize prompt changes for control, augmenting them with more fine-grained control. As part of our experiments, we demonstrate this integration with both Prompt-to-Prompt (P2P) (Hertz et al., 2023) and AdapEdit (Ma et al., 2024), where we simply replace their text modifications with our attribute modulations. Both methods improve consistency with an original generated image when changing the prompt. This combines the benefits of improved disentanglement and structure retainment of these methods with the more fine-grained control of our modulations. We also combine our method with inversion using ReNoise (Garibi et al., 2024) to perform real image editing (see Figure 9). Our combination with AdapEdit uses SD 1.5 (Ma et al., 2024), as AdapEdit is not available for SDXL. Similarly, we use ReNoise with SDXL Turbo (Sauer et al., 2023).

## 4.2 ATTRIBUTE CONTROL FOR IMAGE GENERATION

We evaluate our method's ability to control attribute expression for specific target attributes $A_i$ in different settings and compare it against other approaches both quantitatively and qualitatively. Full descriptions of our experimental setup and evaluation protocols are available in Appendix B.3.

Table 1: Quantitative comparison with other control methods. We evaluate (a) subject-specificity of control in multi-subject settings, (b) disentangledness of attribute control *v.s.* overall image changes, where we normalize the change metrics $\Delta$Id and LPIPS by the attribute expression change $|\Delta\text{CLIP}_{\text{Bi}}|$, (c) whether the method can be used for continuous control, and (d) image generation speed (using an Nvidia A100 80GB SXM at batch size 1).

| Method | (a) Subject-Specificity Subject-Specificity $\uparrow$ | (b) Disentangledness $\Delta$Id $\downarrow$ | LPIPS $\downarrow$ | (c) Continuous | (d) Performance Time $\downarrow$ |
|---|---|---|---|---|---|
| Adjectives in Text Prompt | 4.14 | 0.48 | 0.28 | ✗ | 12.0s [4.17it/s] |
| Concept Sliders (Gandikota et al., 2024) | ✗ | 0.45 | 0.20 | ✓ | 33.8s [1.48it/s] |
| Prompt-to-Prompt (Hertz et al., 2023) | 3.93 | 0.60 | 0.29 | ✗ | 23.5s [4.16it/s] |
| AdapEdit (Ma et al., 2024) | **6.92** | 0.24 | 0.10 | ✗ | 13.2s [7.58it/s] |
| MasaCtrl (Gen.) (Cao et al., 2023) | 2.48 | 0.66 | 0.28 | ✗ | 153.0s [0.65it/s] |
| MasaCtrl (Edit*) (Cao et al., 2023) | 1.93 | 0.61 | 0.43 | ✗ | **10.2s** [4.86it/s] |
| Ours | 3.35 | 0.40 | 0.10 | ✓ | 12.0s [4.17it/s] |
| Ours + Prompt-to-Prompt (Hertz et al., 2023) | 2.23 | 0.37 | 0.08 | ✓ | 23.5s [4.16it/s] |
| Ours + AdapEdit (Ma et al., 2024) | 6.46 | **0.19** | **0.05** | ✓ | 13.2s [7.58it/s] |
| Ours + ReNoise (Garibi et al., 2024) | 2.28 | 0.82 | 0.32 | ✓ | 32.2s [5.367it/s] |
| *Ablations* | | | | | |
| Ours (w/o Delay) | 3.47 | 0.50 | 0.22 | ✓ | 12.0s [4.17it/s] |
| Our CLIP Difference Method (Section 3.2) | 2.38 | 1.20 | 0.58 | ✓ | 12.0s [4.17it/s] |
| Directly modulating $\Delta\hat{\epsilon}$ (Section 3.3) with CFG | 3.15 | 0.73 | 0.39 | ✓ | 23.0s [2.17it/s] |

**best** and 2nd are highlighted. *MasaCtrl editing & AdapEdit are only available for SD 1.5; the other methods use SDXL. CFG denotes Classifier-free Guidance.

**Subject-Specificity of Control** To evaluate subject-specificity, we apply different attribute modulations to individual subjects within multi-subject-prompts. As shown in Figure 12b (see also Appendix A.3 for additional examples), our method can apply attribute modulations independently to each subject $S_j \in \mathcal{S}$ in multi-subject prompts $P$, yielding fine-grained, compositional control. This is despite training the directions $\Delta\mathbf{e}_{A_i}$ only in a single-subject setting. We also find that our modulations enable an extensive coverage of the 2D attribute expression space when applied to multi-subject modulations, improving upon the coverage achieved by other methods (see Figure 11).

For a quantitative evaluation, we use two-subject prompts containing a target entity $S_{\text{target}}$ and another $S_{\text{other}}$ of the same category and measure the change induced by modulating an attribute of one subject relative to the other. Using detected bounding boxes, we calculate the change in CLIP score (a standard metric often used to quantify semantic control magnitudes (Gandikota et al., 2024; Ma et al., 2024)) for both $S_{\text{target}}$ and the other subject $S_{\text{other}}$ as:

$$\Delta\text{CLIP} = 100 \cdot (\text{cossim}_{\text{CLIP}}(\mathcal{I}_{\text{mod}}, P_{\text{edit}}) - \text{cossim}_{\text{CLIP}}(\mathcal{I}_{\text{orig}}, P_{\text{edit}})) \tag{5}$$

where $I_{\text{orig}}$ and $I_{\text{mod}}$ denote the original and edited images, respectively, and $P_{\text{edit}}$ is the desired attribute edit prompt. The cosine similarity $\text{cossim}_{\text{CLIP}}$ measures the alignment between the CLIP embeddings of the images and the attribute-edit prompts. From this, we compute the subject-specificity ratio by comparing the relative change in $\Delta$CLIP for the target subject $S_{\text{target}}$, to the other subject, $S_{\text{other}}$. Formally, we define the subject-specificity metric as:

$$\text{Subject-Specificity} = \frac{|\Delta\text{CLIP}_{(S_{\text{target}})}|}{|\Delta\text{CLIP}_{(S_{\text{other}})}|}. \tag{6}$$

As shown in our evaluation against state-of-the-art control methods in Table 1a, our method retains subject-specificity similar to adding adjectives to the prompt and Prompt-to-Prompt (Hertz et al., 2023), allowing it to achieve fairly isolated changes in attribute expression. AdapEdit, which does not allow continuous modulations, performs substantially better. As AdapEdit uses text prompts to specify changes, we can combine it with our method (unlike other continuous modulation methods such as Concept Sliders, which can not be combined this way) to retain the superior subject-specificity, but also achieve continuous modulations.

**Disentangledness of Control** We also evaluate how disentangled the achieved semantic modulation is from both overall image changes and person identity changes (when applying modulations to people). We quantify overall perceptual image change using LPIPS (Zhang et al., 2018) and for identity similarity, we use the cosine similarity in the ReID embedding space, denoted as $\text{cossim}_{\text{ReID}}$,

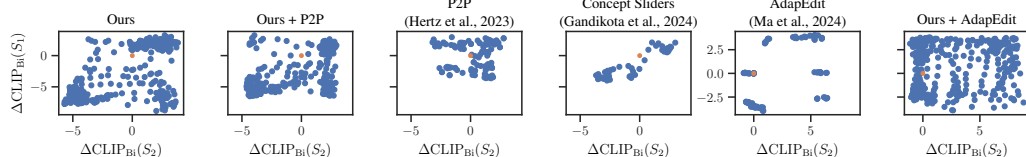

Figure 11: We continuously modulate the target attribute for each of two subjects and estimate the individual attribute expression $\text{expr}_{S_j}(A_i)$ of the target attribute. Our modulations enable reaching a large range of attribute expression combinations, as they are both subject-specific and fully continuous. Other methods are limited in one of these aspects and thus do not allow full coverage. Samples with AdapEdit use SD 1.5, while the rest use SDXL.

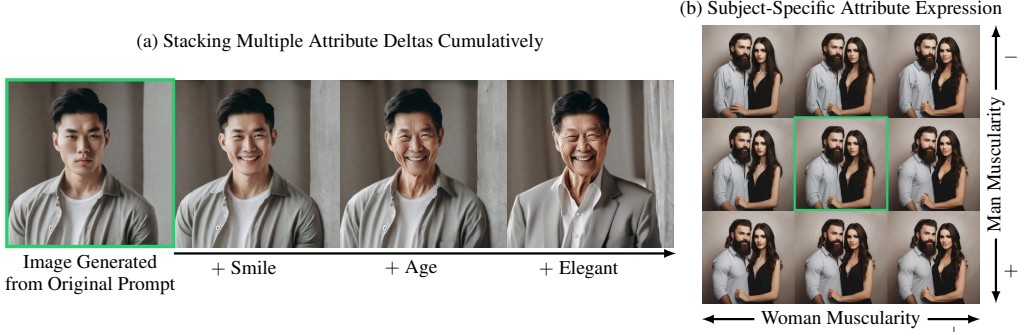

Figure 12: **(a)** Multiple modulations can be composed simply by adding them. **(b)** Modulations can be applied to different subjects with different magnitudes. Unmodified images are marked in **green**.

based on ArcFace embeddings (Deng et al., 2019). The identity change is computed as:

$$\Delta\text{Id} = 1 - \text{cossim}_{\text{ReID}}(\mathcal{I}_{\text{mod}}, \mathcal{I}_{\text{orig}}), \tag{7}$$

We show both results over the magnitude of the achieved semantic change in Figure 13, quantifying the semantic change as a bidirectional CLIP score change:

$$\Delta\text{CLIP}_{\text{Bi}} = \Delta\text{CLIP}_{+} - \Delta\text{CLIP}_{-}, \tag{8}$$

where $\Delta\text{CLIP}_{+}$ uses a positive prompt (e.g., "an old man") and $\Delta\text{CLIP}_{-}$ uses a negative prompt (e.g., "a young man"). This approach enables us to quantify both positive and negative changes in attribute expression faithfully. We also consolidate these results into a single quantitative ratio each for image and person identity change in Table 1b. Compared to other methods, the attribute expression changes achieved with Attribute Control are well-disentangled from auxiliary image changes. When combined with AdapEdit, our method significantly outperforms all other approaches.

**Fine-Grainedness of Control** We further demonstrate the fine-grained control capabilities of our method by showing smooth, gradual modifications in attribute expression across multiple target cat-

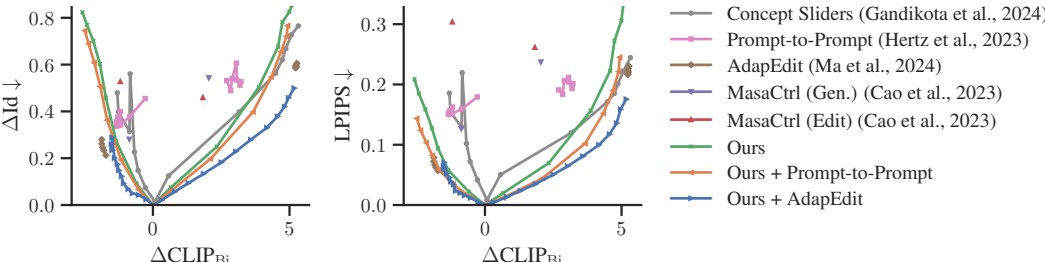

Figure 13: We measure the perceptual change in the image (LPIPS) and the person identity change ($\Delta\text{Id}$) to the unmodified image while modulating the target attribute. Our modulations enable fully continuous and highly disentangled modulations, which is further improved by combining our method with others such as Prompt-to-Prompt or AdapEdit.

egories in Figure 7, qualitatively compared to other methods in Figure 8, and quantitatively evaluated in Figure 13. Unlike other methods such as MasaCtrl (Cao et al., 2023), AdapEdit (Ma et al., 2024), or P2P (Hertz et al., 2023), which do not allow for fine-grained modulations, our approach enables continuous, well-disentangled modulation across a wide range of attribute expression $\mathrm{expr}(A_i)$ similar to Concept Sliders (Gandikota et al., 2024), but while offering subject-specificity. This can also be seen in the multi-subject evaluation in Figure 11.

**Ablation**   We also ablate over different variations of our method (see Table 1). We find that only applying the modulation after the first 20% of steps in the sampling process substantially improves the disentangledness of modulations. Furthermore, we find that our learning-based method for identifying modulation directions significantly improves upon the simple approach introduced in Section 3.2. Similarly, our learned directions are substantially more disentangled than just applying the $\Delta \mathbf{e}$ modulation they were trained on with Classifier-free Guidance (CFG) (Ho & Salimans, 2021) and do not incur the substantial sampling cost overhead.

**Generalization**   We further investigate the generalizability of our method. Generally, any learned modulation direction $\Delta \mathbf{e}_{A_i}$ will have only been trained on a closed set of nouns describing the target subject $S$. To verify that they generalize beyond this set, we apply directions that have been trained on a very small set of generic nouns (e.g., "person", "woman", and "man" for people) to more specific nouns (see Appendix A.2). We find that our directions generalize to this setting as expected. We also find that our learned modulation directions $\Delta \mathbf{e}_{A_i}$ can generalize to other models that use the same text encoders in a *zero-shot* manner. By learning a direction on one model, in this case, SDXL (Podell et al., 2024), we can directly transfer it to models that use the same text encoders (see Figure 10), such as SDXL Turbo (Sauer et al., 2023), or a subset of them, as with SD 1.5 (Rombach et al., 2022) or the image+depth model LDM3D (Stan et al., 2023). Our learned directions even generalize to non-diffusion models such as aMUSEd (Patil et al., 2024).

## 5   CONCLUSION

This work uncovers the powerful capabilities of the tokenwise CLIP Radford et al. (2021) text embedding for exerting control over the image generation process in T2I diffusion models. Instead of just acting as a discrete space of embeddings of words, we find that diffusion models are capable of interpreting local deviations in the tokenwise CLIP text embedding space in semantically meaningful ways. We use this insight to augment the typically rather coarse prompt with fine-grained, continuous control over the attribute expression of specific subjects by identifying semantic directions that correspond to specific attributes. Since we only modify the tokenwise CLIP text embedding along pre-identified directions, we enable more fine-grained manipulation at no additional cost in the generation process.

**Limitations and Future Work**   This work is a step towards revealing the hidden capabilities of the text embedding input to common large-scale diffusion models and making them usable in straightforward ways. While our approach works for different off-the-shelf models without modifying them, it is also inherently limited by their capabilities. Specifically, our method inherits the limitation that diffusion models sometimes mix up attributes between different subjects. Complementary methods (Chefer et al., 2023; Rassin et al., 2023) reduce these problems substantially, and future work could investigate their combination with our method in depth. Our approach also uses linear modulations along semantic directions in CLIP's tokenwise embedding space. In GANs, where similar linear modulations are often used, previous works (Balakrishnan et al., 2022) found that more disentangled changes can be achieved using nonlinear modulations. The tokenwise CLIP text embedding space might share this property and could benefit from applying similar strategies to further improve disentanglement.

ETHICS STATEMENT

This work aims to improve the capabilities of text-to-image diffusion models by enabling more fine-grained control over generated content, with applications to controlled generation and image editing. Text-to-image models can generally be used to create misleading or inappropriate content and may inherit biases from training data, including gender, race, and cultural stereotypes. Our method offers a potential mitigation strategy for some of these issues, helping to counteract biases by providing users with more precise control over generated images instead of purely relying on the pre-trained model to determine appropriate attribute combinations. We encourage responsible use and further research into mitigating biases in text-to-image generation.

REPRODUCIBILITY STATEMENT

In addition to the information given in the main body of the paper, we provide extensive details about both our method (Appendices B.1 and B.2) and experiments (Appendix B.3), including implementation considerations and hardware, in the appendix. Further, we provide a fully documented reference implementation of our method in the supplementary material. For figures, we also include additional details like prompts used for generated images in Appendix C.

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

CONTENTS

# A    ADDITIONAL EXPERIMENTS & RESULTS

## A.1    POSTFIX ATTRIBUTE LEARNING

Some attributes are not easily expressible as prefixes to the noun. This means that, due to the causal nature of the CLIP text encoder, our optimization-free method for identifying attribute directions (see Section 3.2) can not be applied. However, we find that this limitation does not apply to our optimization-based approach (see Section 3.3): we can learn directions based on attributes expressed as postfixes (e.g., *"a person* *wearing sunglasses**"*, for which we show a qualitative example in Figure 14).

*+ "... wearing sunglasses"*

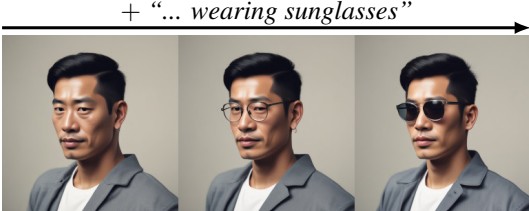

Figure 14: Our learning-based method can also learn to represent attributes represented as postfixes to the target subject noun during training.

## A.2 SUBJECT NOUN TRANSFERABILITY

We investigate how much our learned attribute modulations can generalize across different nouns that describe the same subject. We generally learn them on a set of different nouns that describe a subject of a specific category (e.g., for people with the words "man", "woman", and "person"). However, these words typically do not cover the whole range of possible nouns that can be used to describe subjects of a general category. Ideally, one could learn one modulation for one concept, such as age, on a small set of nouns and generalize across all nouns of a category or even to subjects of other categories.

First, we test the generalization of modulations learned for people on "man", "woman", and "person" and apply them to increasingly more specific nouns that describe people. Results are shown in Figures 15 and 16, and all prompts are "a photo of a beautiful <noun>". As a baseline, we apply them to "child", "mother", and "father", three words that are previously unseen but still describe very high-level sub-categories of people. We find that the learned modulations still work as expected. Similarly, for categories of jobs such as "doctor", "barista", or "firefighter", which are substantially more specific and also substantially affect their clothing and the rest of the image, we find that they also work well. Finally, applying these learned modulations to very specific nouns such as the names "John" and "Jane" also works as expected. This demonstrates that our learned modulations can generalize well across a wide range of unseen nouns describing instances of a specific category, even if they were only learned on a small set of high-level, potential nouns.

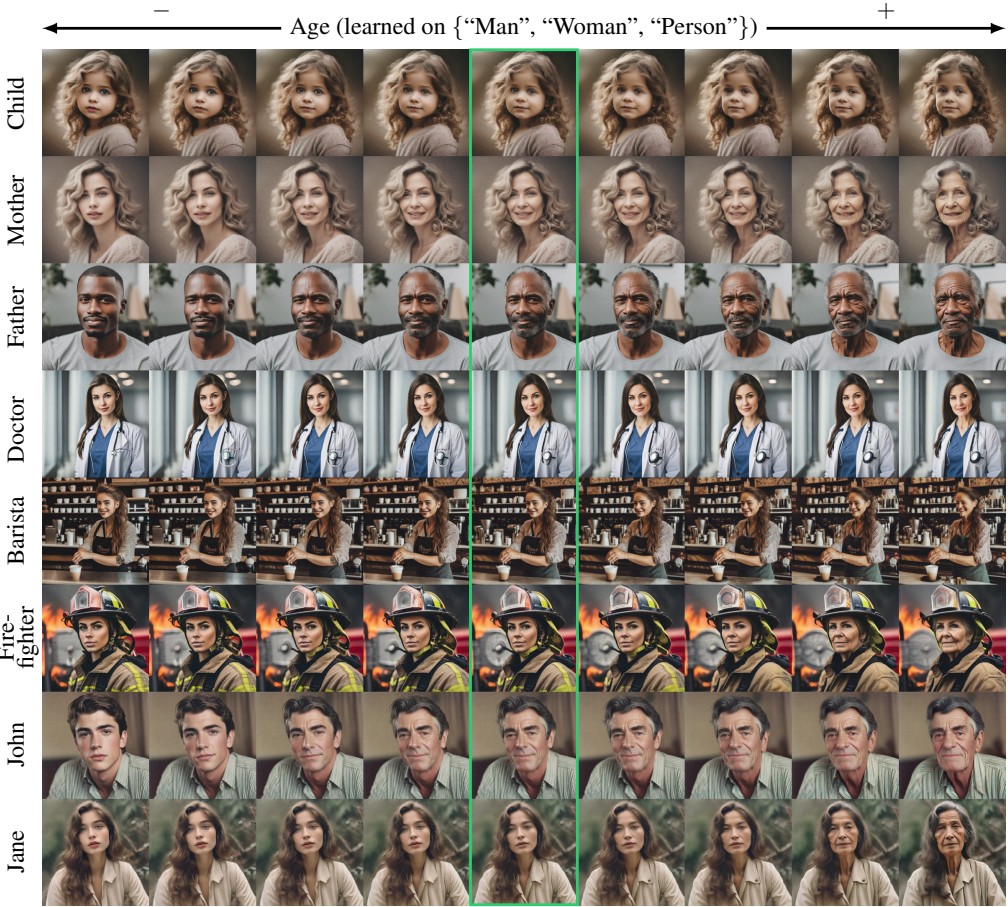

Figure 15: **Subject Noun Transferability**. We stress-test applying modulations that have been learned only on the nouns "man", "woman", and "person" to various other nouns that describe people. The unmodified image is marked in **green**. All samples are generated using attribute modulations being applied with a linear scale from -2 to 2 across each.

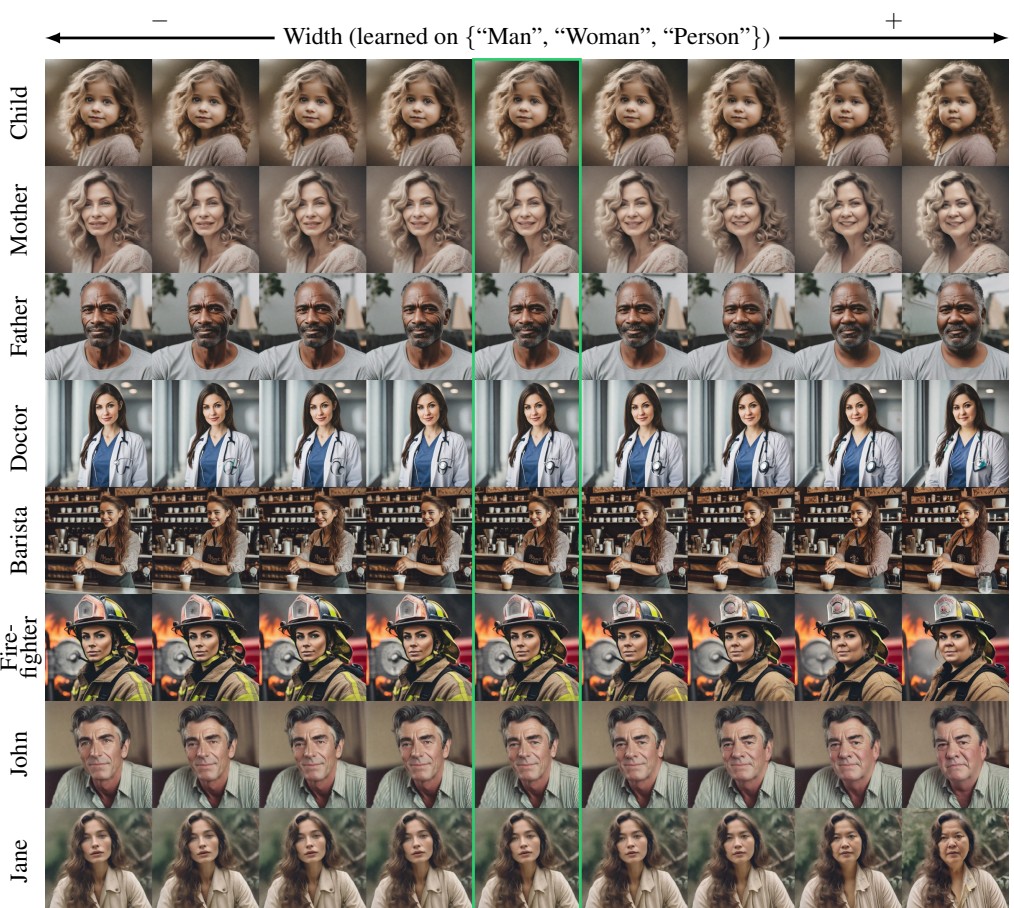

Figure 16: **Subject Noun Transferability**. We stress-test applying modulations that have been learned only on the nouns "man", "woman", and "person" to various other nouns that describe people. The unmodified image is marked in **green**. All samples are generated using attribute modulations being applied with a linear scale from -2 to 2 across each.

## A.3 MULTI-SUBJECT ATTRIBUTE EDITING

Figures 17 and 18 show examples of modulating attributes in a subject-specific manner using our learned modulations. These show that various attributes can be applied to subjects individually, even if both subjects are of the same category (e.g., "people"). A slight correlation between, e.g., the age of the man and the age of the woman in Figure 17 is visible and expected, as the diffusion model also models these dependencies between different subjects in the generated image. By applying both modulations with different strengths, the whole spectrum of combinations can be achieved, as shown in Figure 11.

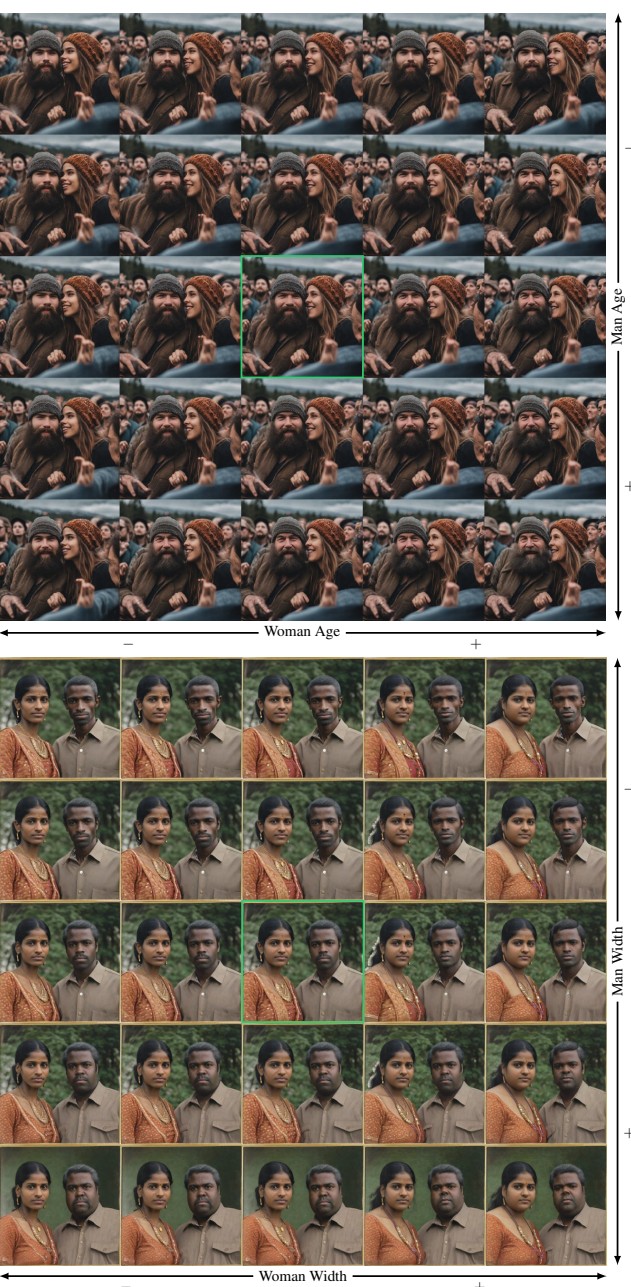

Figure 17: **Multi-Subject Attribute Modifications**. The unmodified image is marked in green. All samples are generated using one attribute modulation each being applied to the two subjects mentioned in the prompt with a linear scale from -2 to 2 across each.

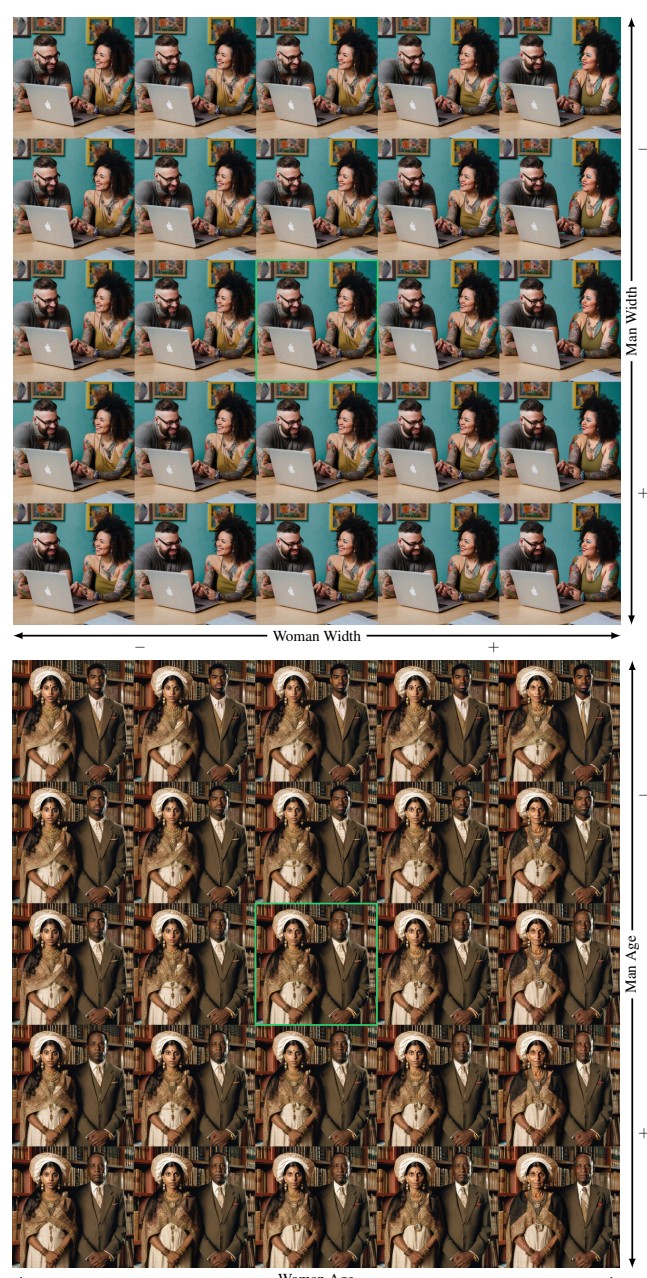

Figure 18: **Multi-Subject Attribute Modifications**. The unmodified image is marked in **green**. All samples are generated using one attribute modulation each being applied to the two subjects mentioned in the prompt with a linear scale from -2 to 2 across each.

## A.4 COMPOSITIONAL ATTRIBUTE EDITING

We show some 2d grids where two attributes are modulated for the same target subject in an additive manner in Figures 19 and 20. Both attribute modulations interact with each other according to the world knowledge of the diffusion model to produce a realistic image for every combination.

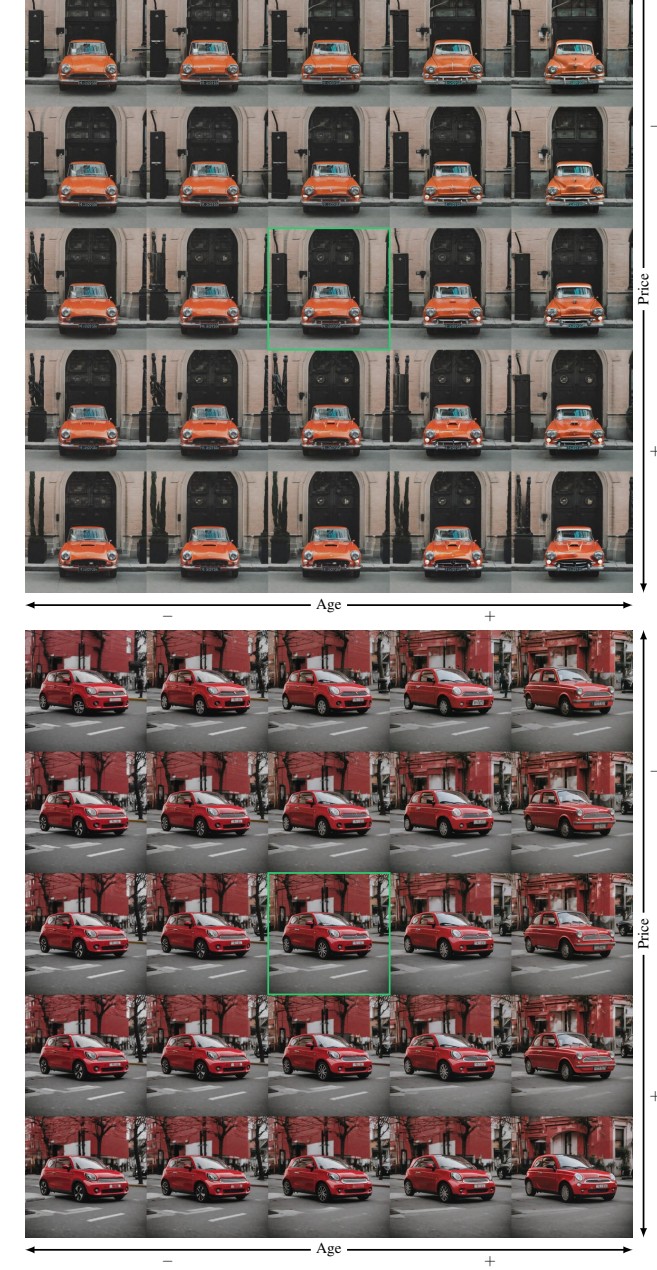

Figure 19: **Compositional Attribute Modifications**. The unmodified image is marked in green. All samples are generated using two attribute modulations being applied additively with a linear scale from -2 to 2 across each.

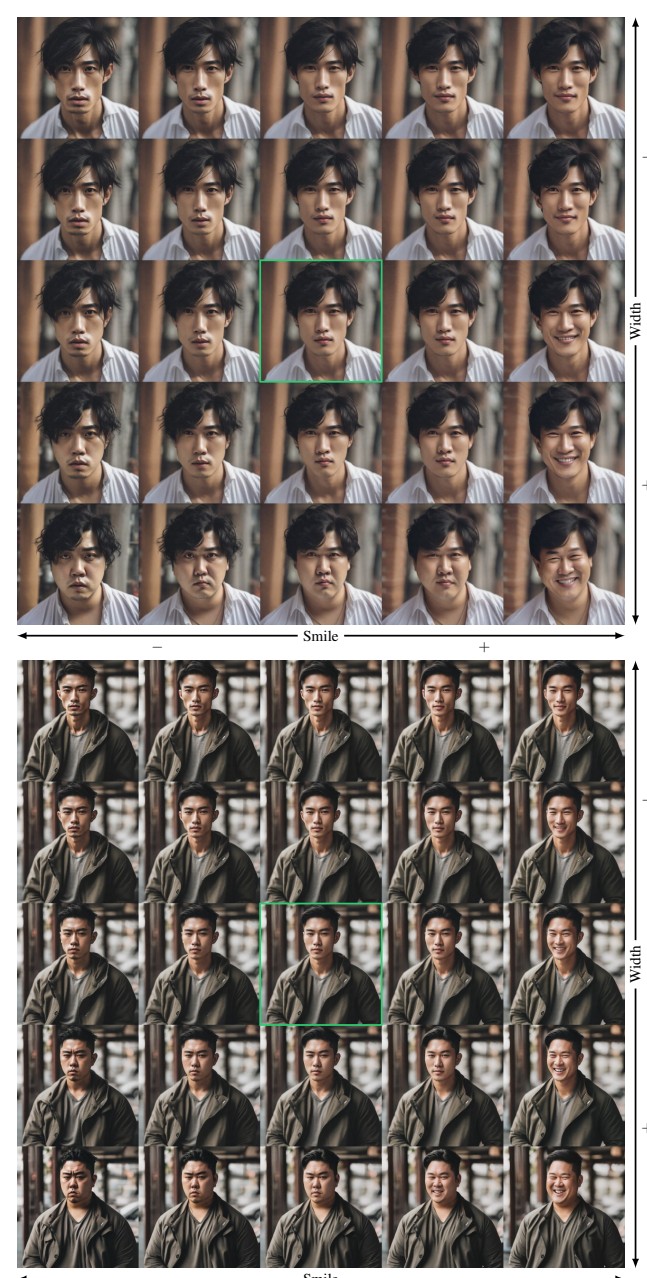

Figure 20: **Compositional Attribute Modifications**. The unmodified image is marked in **green**. All samples are generated using two attribute modulations being applied additively with a linear scale from -2 to 2 across each.

## A.5 CONTINUOUS ATTRIBUTE MODULATION

To illustrate the breadth of attributes that can be modulated and how continuous the attribute changes are, we show a range of attributes being continuously modulated. Figures 21 to 24 show examples where attribute modulations are applied with our delayed sampling, Figure 25 shows attribute modulations applied for the full sampling time. For every category, we re-use the same sample instances as a starting point.

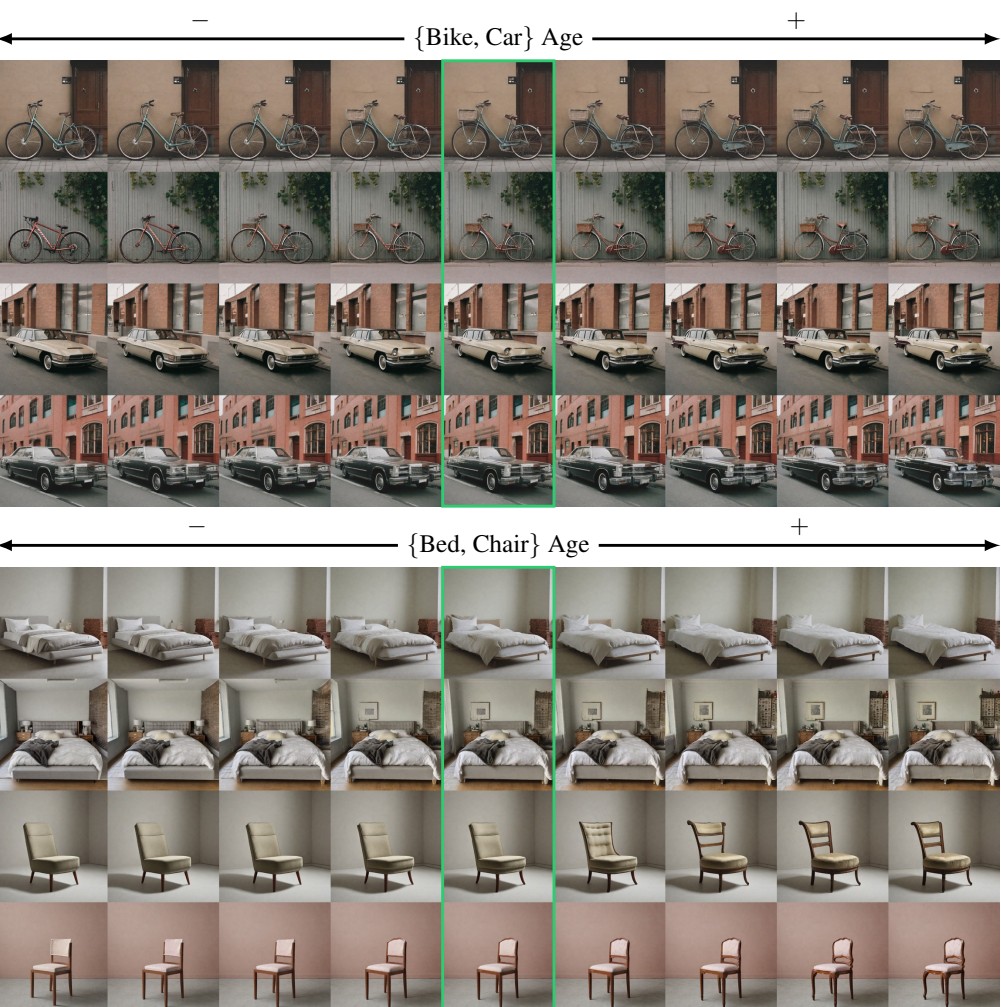

Figure 21: **Continuous Attribute Modifications**. Unmodified images are marked in **green**. All samples are generated using a linear scale from -2 to 2.

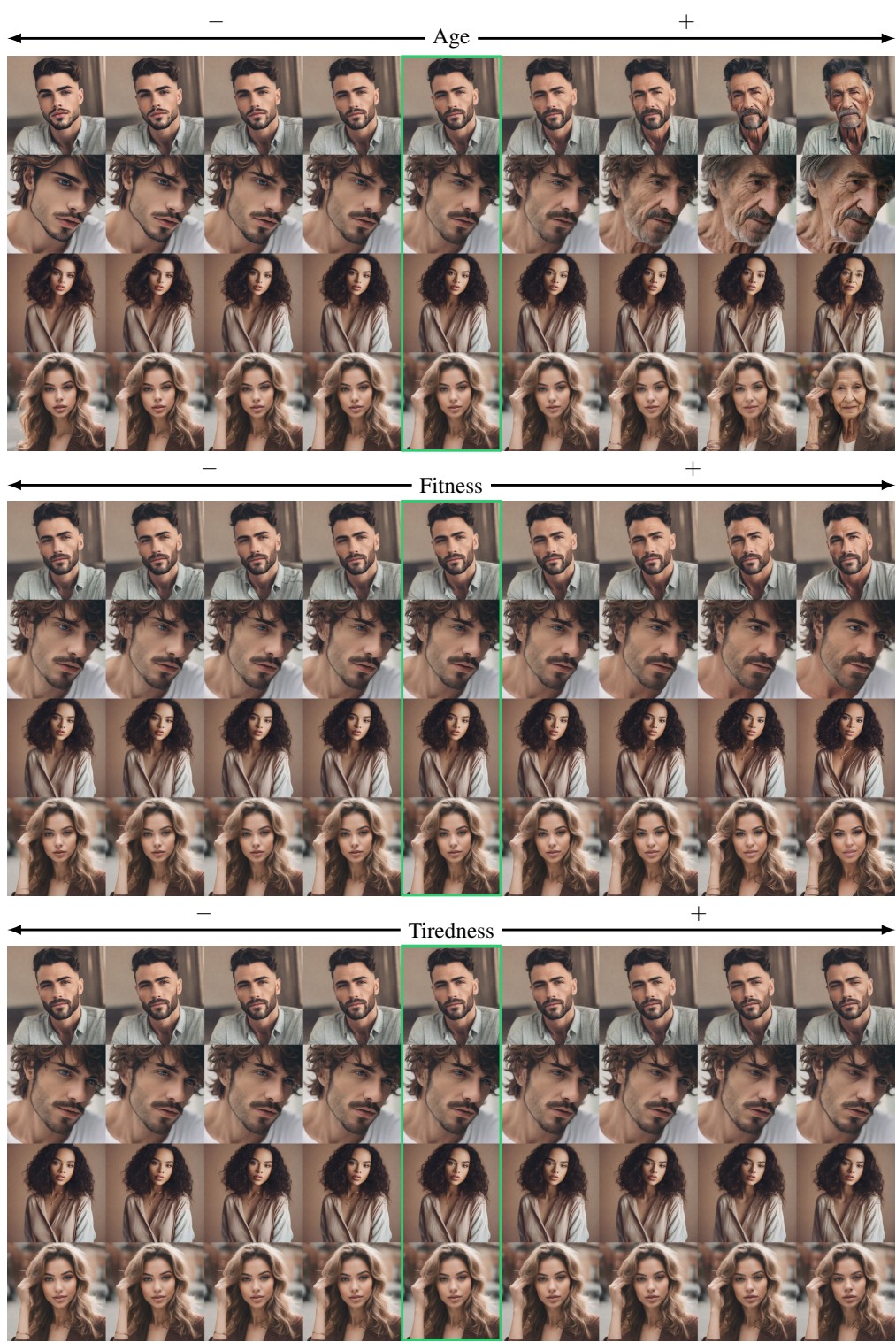

Figure 22: **Continuous Attribute Modifications**. Unmodified images are marked in green. All samples are generated using a linear scale from -2 to 2.

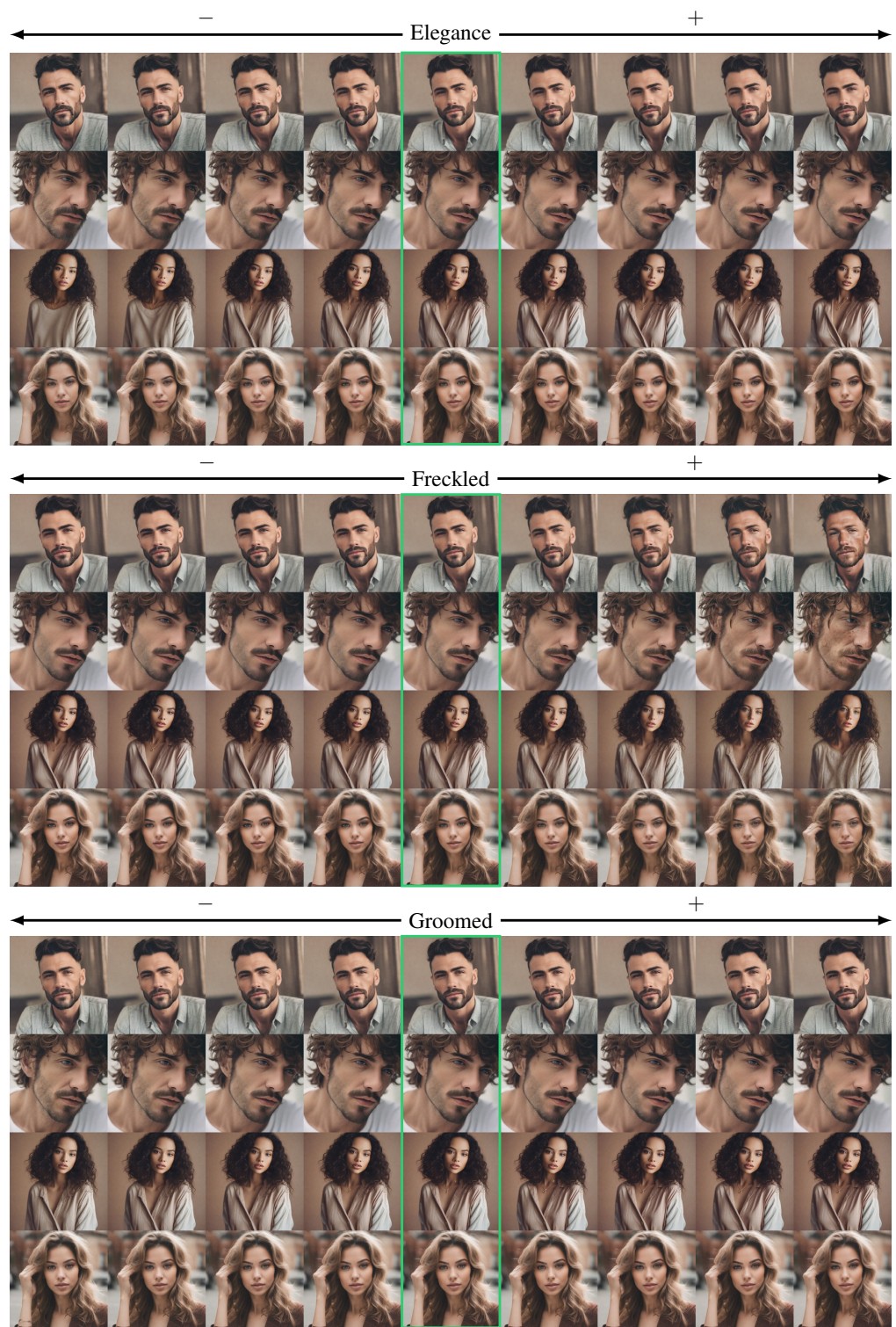

Figure 23: **Continuous Attribute Modifications**. Unmodified images are marked in green. All samples are generated using a linear scale from -2 to 2.

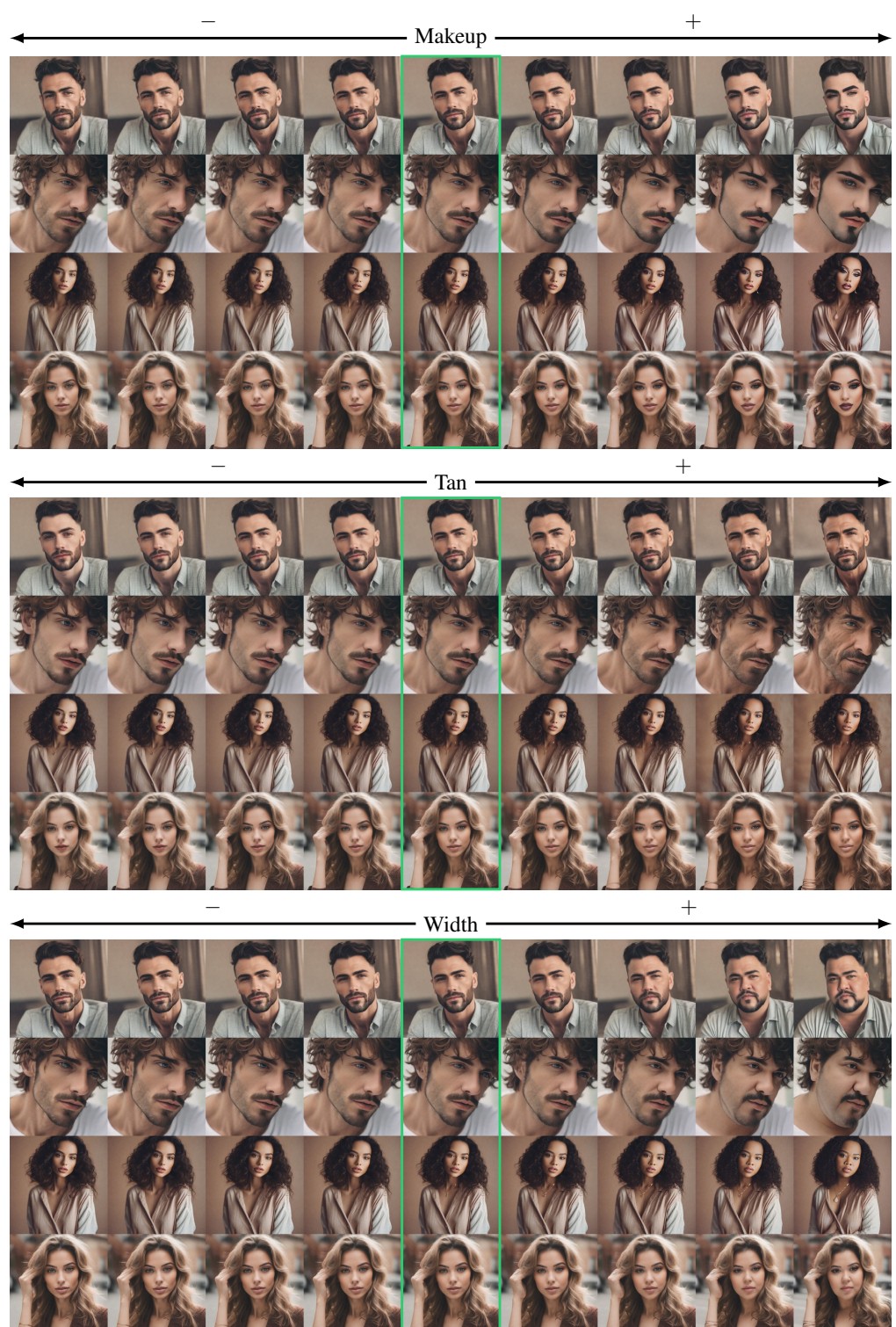

Figure 24: **Continuous Attribute Modifications**. Unmodified images are marked in **green**. All samples are generated using a linear scale from -2 to 2.

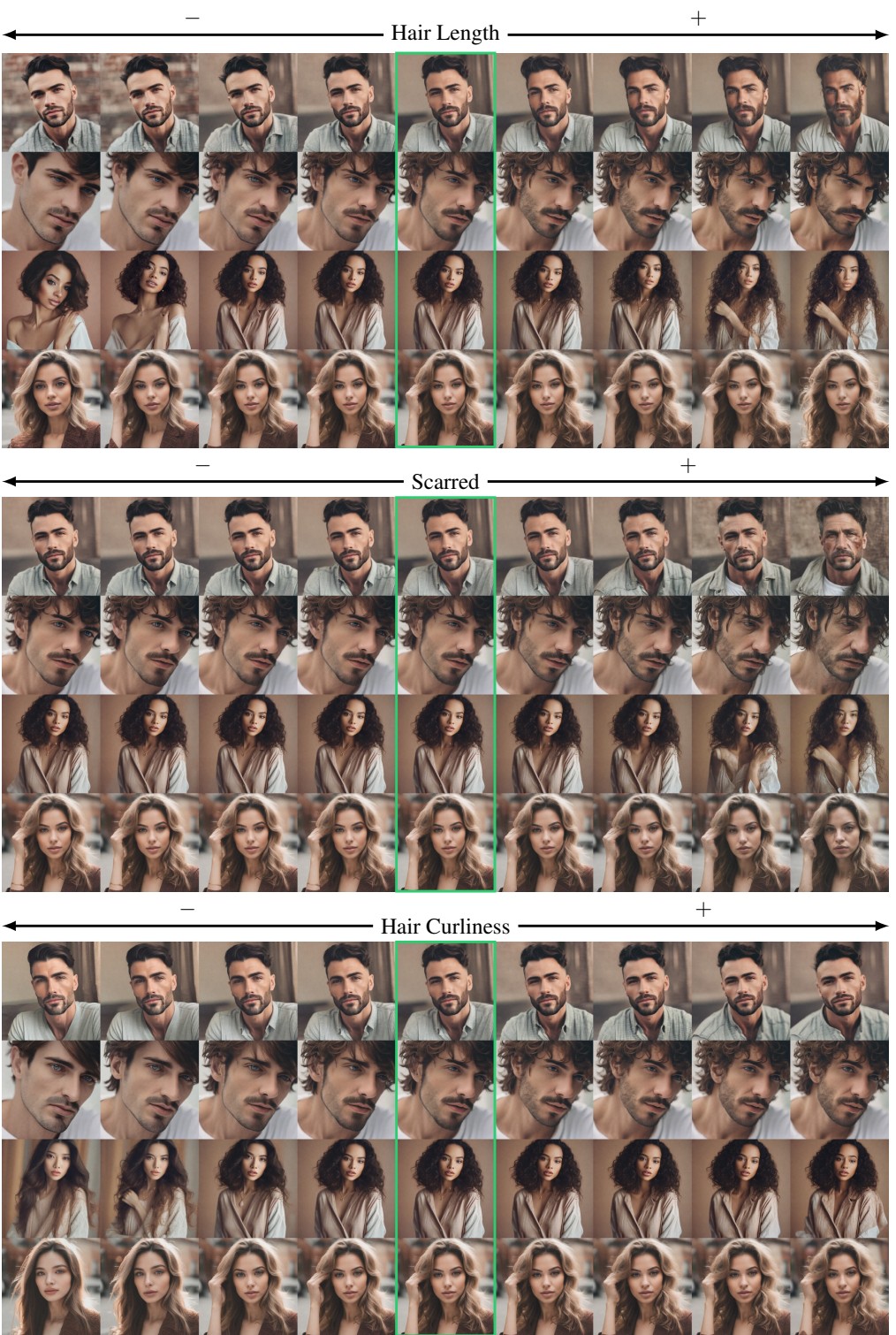

Figure 25: **Continuous Attribute Modifications**. Unmodified images are marked in green. All samples are generated using a linear scale from -2 to 2, with the modulations being applied for all steps (w/o Delay).

## B  IMPLEMENTATION DETAILS

This section gives details about the implementation of our method. We generally use the default settings as set in `diffusers`[2]-v0.25.0 with a classifier-free guidance (Ho & Salimans, 2021) scale of 7.5 and 50-step DDIM (Song et al., 2021) sampling unless specified otherwise.

### B.1  SEMANTIC DIRECTION TRAINING

---

**Algorithm 1** Algorithm for Learning the Semantic Directions

---

1: **Input:**
  Pre-trained diffusion model $\hat{\epsilon}_\theta$
  CLIP embedding dimension $d_{\text{CLIP}}$
  Learning rate $\eta$, number of steps $S$, batch size $B$
2: **Output:**
  Learned semantic direction $\Delta \mathbf{e}_{A_i}$
3: Initialize $\Delta \mathbf{e}_{A_i} = \mathbf{0}$                  ▷ Initialization
4: **for** $s = 1$ to $S$ **do**                   ▷ Training loop
5:   $\mathcal{L}_{\text{batch}} \leftarrow 0$                ▷ Initialize batch loss
6:   **for** each entry in batch of size $B$ **do**
7:     Sample random subject $S_j$ and neutral prompt $P$
8:     Generate image $\mathbf{x}_0$ from neutral prompt $P$
9:     $t \sim \mathcal{U}[0, T]$           ▷ Sample random timestep
10:     $\mathbf{x}_t = \alpha_t \mathbf{x}_0 + \sigma_t \boldsymbol{\epsilon}, \boldsymbol{\epsilon} \sim \mathcal{N}(0, \mathbf{I})$       ▷ Add noise
11:     $\tilde{\boldsymbol{\epsilon}} = \hat{\epsilon}_\theta(\mathbf{x}_t|P)$         ▷ Predict noise for $P$
12:     $\tilde{\boldsymbol{\epsilon}}_+ = \hat{\epsilon}_\theta(\mathbf{x}_t|P_+)$        ▷ Predict noise for $P_+$
13:     $\Delta\tilde{\boldsymbol{\epsilon}} = \tilde{\boldsymbol{\epsilon}}_+ - \tilde{\boldsymbol{\epsilon}}$       ▷ Compute noise direction
14:     $\lambda_i \sim \mathcal{U}([-5, 5] \setminus (-0.1, 0.1))$     ▷ Sample scale factor
15:     $\mathcal{L}_i = w(t) \|(\boldsymbol{\epsilon} + \lambda_i \Delta\tilde{\boldsymbol{\epsilon}}) - \hat{\epsilon}_\theta(\mathbf{x}_t|\mathbf{e}'(\mathbf{e}, \lambda_i \Delta \mathbf{e}_{A_i}), t)\|_2^2$   ▷ Compute loss for this entry
16:     $\mathcal{L}_{\text{batch}} \leftarrow \mathcal{L}_{\text{batch}} + \mathcal{L}_i$      ▷ Accumulate batch loss
17:   **end for**
18:   Compute mean loss for the batch: $\mathcal{L}_{\text{mean}} \leftarrow \frac{1}{B}\mathcal{L}_{\text{batch}}$
19:   Update $\Delta \mathbf{e}_{A_i}$ using AdamW optimizer with learning rate $\eta$ based on $\mathcal{L}_{\text{mean}}$
20: **end for**
21: **Return:** $\Delta \mathbf{e}_{A_i}$

---

The semantic directions $\Delta \mathbf{e}_{A_i}$ for target attribute $A_i$ are implemented as learnable parameters of shape $1 \times d_{\text{CLIP}}$, with $d_{\text{CLIP}}$ being the embedding dimension of the CLIP text encoder. For SDXL (Podell et al., 2024), this is 2048, resulting from the channelwise concatenation of embeddings from the OpenAI CLIP ViT-L (Radford et al., 2021) and OpenCLIP ViT-bigG (Ilharco et al., 2021). This direction is applied additively with scaling according to Equation (3) to the target subject tokens (e.g., "person" in the case of "a photo of a person") in the original text embedding $\mathbf{e}$. If the target subject consists of multiple tokens, we broadcast $\Delta \mathbf{e}_{A_i}$ across those tokens, although this is only very rarely the case in practice. Similarly, if one subject is mentioned in the prompt multiple times, we apply the same modulation to all instances.

We train our semantic directions $\Delta \mathbf{e}_{A_i}$ for 1000 steps[3] at a batch size of 10. We use AdamW (Loshchilov & Hutter, 2019) with a learning rate of 0.1, $(\beta_1, \beta_2) = (0.5, 0.8)$, and weight decay of 0.333. All directions are trained on a single A100 with 40GB of VRAM using a bfloat16 version of SDXL (Podell et al., 2024).

For every entry in the batch, we use a random combination of prefix prompt (e.g. "an photo of", optionally with attributes such as ethnicity, to focus the implied direction on one that is invariant to these attributes) and prompt tuple (e.g "a woman") and sample an image with the neutral prompt (e.g. ("a photo of a woman") and a random seed, stopping at a random timestep. We then compute

---

[2] https://github.com/huggingface/diffusers
[3] The directions tend to be mostly converged after 10 steps, but we train for a unified training time across all attributes for consistency.

the prediction starting from that step for all two/three prompts, resulting in $\tilde{\epsilon}, \tilde{\epsilon}_+$, and optionally $\tilde{\epsilon}_-$. In contrast to Gandikota et al. (2024), we explicitly distill the full direction implied by $\Delta\tilde{\epsilon}$ by using multiple scales $\lambda_i$ sampled from a continuous scale distribution. Preliminary experiments showed that this helps obtain substantially more robust directions. Additionally, we sample our starting samples using standard sampling instead of a modified generation process.

We then sample four values for $\lambda_i \sim \mathcal{U}([-5, 5] \setminus (-0.1, 0.1))$ and compute our training loss (Equation (4)) over them. We found that sampling multiple values for $\lambda_i$ substantially boosts the quality of our learned directions at little overhead cost (as the online sampling of the original images is the most costly part) and that values for $\lambda_i$ very close to zero were not particularly useful for the training process. Empirically, we find that most of our learned directions are already close to convergence after five optimization steps, but we keep training for the full time for simplicity.

## B.2 COMBINATION OF ATTRIBUTE CONTROL WITH OTHER METHODS

In Section 4, we combine our attribute control method with other off-the-shelf controlled generation methods.

**Combination with Prompt-to-Prompt (Hertz et al., 2023)** To combine our method with Prompt-to-Prompt, we apply the standard Prompt-to-Prompt method. We use the same adaptation mode and hyperparameters as used for adding adjectives in the text prompt, but add our modulations on the text prompt embedding instead. To modulate the change, we scale our directions as usual.

**Combination with AdapEdit (Ma et al., 2024)** AdapEdit uses the same general external interface as Prompt-to-Prompt. Here, we apply our modulations in the exact same way as previously described for Prompt-to-Prompt. As AdapEdit is not available for SDXL (Podell et al., 2024), we use zero-shot adaptation of our semantic directions obtained on SDXL to SD1.5, as described in Section 4.2.

**Combination with ReNoise (Garibi et al., 2024)** To apply our controlled generation approach to editing, we combine it with ReNoise, a standard inversion approach. We use their official reference implementation based on SDXL Turbo (Sauer et al., 2023) and apply our modulations learned on SDXL there. We perform inversion purely with ReNoise with default settings and an image description prompt to obtain a starting latent $\mathbf{x}_T$, and then perform controlled generation purely with our method with standard settings. This could optionally be combined further with other methods during inference, such as Prompt-to-Prompt (Hertz et al., 2023) and AdapEdit (Ma et al., 2024).

## B.3 EXPERIMENT EVALUATION DETAILS

To compute perceptual image differences, we use LPIPS (Zhang et al., 2018) as implemented in the `lpips`[4] package with default settings at a resolution of $256^2$ (interpolated bi-linearly). For CLIP scores, we use the standard implementation in `torchmetrics`[5] (which outputs cosine similarities scaled to $[0, 100]$) with default settings, including the default CLIP choice of the CLIP-ViT-L/14 trained by OpenAI (Radford et al., 2021). For image-image similarity evaluations with DINOv2 (Oquab et al., 2024), we use the ViT-L/14 variant with registers (Darcet et al., 2024) and bi-linearly resize to $224^2$ before passing them to the model and comparing the cosine similarity of the CLS token outputs. Finally, for ReID evaluations, we use the ArcFace (Deng et al., 2019) implementation provided by the `insightface`[6] python package with the default `buffalo_l` model, where we compute the cosine similarity of the embeddings of the detected faces.

**Implementations of other Methods** For Concept Sliders (Gandikota et al., 2024), we use the official public implementation[7]. For Prompt-to-Prompt (Hertz et al., 2023), we use the unofficial port of the method to Stable Diffusion XL[8]. This implementation also served as the basis for integrating our method with Prompt-to-Prompt in our codebase. As this implementation is partially incomplete,

---

[4] https://github.com/richzhang/PerceptualSimilarity
[5] https://github.com/Lightning-AI/torchmetrics
[6] https://github.com/deepinsight/insightface
[7] https://github.com/rohitgandikota/sliders
[8] https://github.com/RoyiRa/prompt-to-prompt-with-sdxl

we referred to the official implementation[9] for the implementation of reweighting of added words. For AdapEdit[10], MasaCtrl[11], and ReNoise[12], we also used the respective official implementations. When comparing attribute modulation capabilities across different methods, we compare using the target attribute age on people, as this attribute is i) unambiguous in what exactly it describes, ii) relatively well objectively quantifiable unlike the vast majority of attributes, iii) fully continuous, and iv) the only reasonable attribute that is supported by Concept Sliders[13].

**Attribute Distribution Shifts (Figure 6)**    For each value of $\lambda_i \in \{0, 1, 2, 3\}$, 20 samples (with fixed seeds across scales) were drawn. We compute the delta CLIP score as specified in the experiments section of the paper and use scipy's Gaussian KDE method[14] to compute the kernel density estimate for the resulting distributions with Scott's rule and default settings.

**Qualitative Continuous Modulation (Figure 8)**    We continuously modulate the age of the person described in the prompt with both our method and Concept Sliders (Gandikota et al., 2024), choosing coefficients such that a wide range is covered and both methods show similar scales per column. For Prompt-to-Prompt (Hertz et al., 2023) and MasaCtrl (Cao et al., 2023), we add "old" or "young" to the prompt to coarsely modulate the target attribute. Prompt-to-Prompt further enables some fine-grained control *around the already offset attribute expression point from the added adjective* by re-weighting the added adjective. This does, at least for Stable Diffusion XL (Podell et al., 2024), not allow continuous modulation back to the original image, causing a discontinuity. This can intuitively be explained by the fact that attributes are aggregated in the subject noun, a fact that our method exploits to directly enable fine-grained, subject-specific target attribute modulation: as the attribute modulation for P2P is already partially contained in the subject noun, modulating just the added adjective's cross-attention map can not fully recover the original generated image. At the same time, when combined with our method, where we just modulate the target subject noun's embedding instead of adding new adjectives, this problem immediately subsides.

**Quantitative Subject Specificity Evaluation (Table 1a)**    With each method, we generate variations across a set of 50 images with individual prompts describing two people, where we modulate the target attribute of one of the two subjects. We detect each subject in the unmodified image as previously described with the standard pipeline from `insightface`, and then compute the target metric for each bounding box. We aggregate the specificity metric as described in Equation (6) by computing the fraction individually per sample and then aggregating the overall mean. As there are some cases where this effectively results in a division by zero, we clamp the resulting individual values to $[0, 10]$. We chose 10 as a threshold, as it prevents these outlier samples from having an extraordinarily strong effect on the overall mean.

**Attribute Coverage Evaluation (Figure 11)**    To evaluate the set of attribute combinations reachable by each method, we start from the same setup as previously described for Table 1a, but continuously modulate the age for both subjects visible in the image, covering all combinations of modulation scales for each method. We evaluate 20 values per subject, producing 400 generated samples per method for methods that allow independent continuous modulation of both subjects. We then measure the attribute expression for each subject bounding box (obtained as previously in Table 1a) using Equation (8) and plot the distribution for one representative sample in Figure 11.

**Quantitative Disentangledness Evaluation (Figure 13, Table 1b)**    We generate 50 base samples showing people with different prompts of the format *"a close-up portrait of a {modifiers} {woman, man}"*, where {modifiers} describes a set of prefixes (e.g., *"{∅, beautiful, elegant} asian"*, *"{∅, beautiful, elegant} african-american"*, etc) to cover a wide variety of different images. Then, we modulate the target attribute continuously using each method. We then measure the attribute expression change with Equation (8), the image change with LPIPS, and the identity change as in

---

[9]`https://github.com/google/prompt-to-prompt`
[10]`https://github.com/AnonymousPony/adap-edit`
[11]`https://github.com/TencentARC/MasaCtrl`
[12]`https://github.com/garibida/ReNoise-Inversion`
[13]`https://sliders.baulab.info/weights/xl_sliders/`
[14]`https://docs.scipy.org/doc/scipy/reference/generated/scipy.stats.gaussian_kde.html`

Equation (7). We aggregate these values over all 50 images per combination of method & hyperparameters and then plot them in Figure 13. For $Table\ 1b$, we compute the slope of these graphs (using the absolute value of $\Delta\text{CLIP}_{\text{Bi}}$ for the denominator, to account for the fact that the changes increase for positive values and one for negative values of $\Delta\text{CLIP}_{\text{Bi}}$) to quantify the disentangledness of the edits both from overall visual changes (LPIPS) and person identity changes ($\Delta\text{Id}$).

**Inference Performance Evaluation (Table 1d)** For each method, we use the released implementations of each respective method with default settings and replicate the original environments as closely as possible, given the information documented by the authors. We measure inference times on the same Nvidia A100 SXM with 80GB of RAM and document both the total time and (average) step time, as some methods use different step counts for sampling. For the main paper, we consolidate inversion and generation time if applicable.

## C   VISUALIZATION DETAILS & PROMPTS

Generally, all examples in the paper use Stable Diffusion XL as introduced in Podell et al. (2024) unless noted otherwise.

**Figure 1**   Prompt: *"A close-up photo of a man and a woman sitting on a bench."*

**Figure 2**   Prompts: *"a portrait of a beautiful car"*, *"a portrait of a beautiful frog"*, and *"a portrait of a beautiful suv"*.

**Figure 3**   Prompt: *"a portrait of a beautiful woman with her beautiful dog"*.

**Figure 4**   Prompt: *"a photo of a car"*.

**Figure 6**   Prompt: *"a photo of a car"*.

**Figure 7**   Prompt 1: *"a portrait of a beautiful chair"*.
Prompt 2: *" photo of an old car"*.
Prompt 3: *"a portrait of a beautiful truck"*.
Prompt 4: *"a photo of a beautiful man"*.

**Figure 8**   Base prompt: *"a close-up portrait of a indian woman"*.

**Figure 10**   aMUSEd: *"a photo of a beautiful man"*.
SD 1.5: *"a headshot of a relaxed woman and a friendly man"*.

**Figure 9**   Image 1 is a photo with the title *"a red rolls royce parked in front of a building"* by Rico Reynaldi, obtained from Unsplash[15]. The image is licensed under the Unsplash license[16] and has been center-cropped for inversion.
Inversion Prompt: *"a photo of a beautiful red car on the top deck of a parking garage with large buildings in the background, hazy weather with sunshine"*.
Image 2 is a photo by The Royal Society, obtained from Wikimedia[17]. The image is licensed under the Creative Commons Attribution-Share Alike 3.0 Unported license[18] and has been cropped to primarily show the person's head.
Inversion Prompt: *"a photo of a man wearing glasses and a suit"*.

---

[15]https://unsplash.com/photos/a-red-rolls-royce-parked-in-front-of-a-building-sAN11DGnjqk
[16]https://unsplash.com/license
[17]https://commons.wikimedia.org/wiki/File:Demis_Hassabis_Royal_Society.jpg
[18]https://creativecommons.org/licenses/by-sa/3.0/deed.en

**Figure 12a**   Prompt: *"a photo of a beautiful asian man"*.

**Figure 12b**   Prompt: *"a portrait of a bearded man and a beautiful brunette woman"*.

**Figures 15 and 16**   Prompt Template: *"a photo of a beautiful [...]"*

**Figure 17**   Prompt 1: *"a photo of a bearded man in a beanie enjoying a concert with a bohemian woman in flowing attire"* Prompt 2: *"a portrait of an indian woman standing next to an african-american man"*

**Figure 18**   Prompt 1: *"a photo of a tech-savvy man with a laptop engaged in conversation with a creative woman with colorful tattoos"* Prompt 2: *"a portrait of an indian woman dressed in traditional clothing next to an african-american man wearing a hat standing in a library"*

**Figure 19**   Prompt 1: *"a photo of a car"* Prompt 2: *"a photo of a compact red car"*

**Figure 20**   Prompt 1 & 2: *"a photo of a beautiful asian man"*

**Figure 21**   Prompt 1 & 2: *"a photo of a bike"*
Prompt 3 & 4: *"a photo of a car"*
Prompt 5 & 6: *"a photo of a bed"*
Prompt 7 & 8: *"a photo of a chair"*

**Figures 22 to 25**   Prompt 1 & 3: *"a photo of a beautiful man"*
Prompt 2 & 4: *"a photo of a beautiful woman"*

