# OpenReview forum: "Continuous, Subject-Specific Attribute Control in T2I Models by Identifying Semantic Directions"
_ICLR.cc/2025/Conference — ICLR 2025 Conference Withdrawn Submission_

### Official Review · Reviewer_66f7 · 2024-10-18

**Soundness:** 2
**Presentation:** 3
**Contribution:** 2
**Rating:** 3
**Confidence:** 3

**Summary:**

The authors proposed a method to achieve fine-grained control over the attributes of images generated by diffusion models. They suggested introducing an adjective into the prompt and then comparing the embedding distances to derive a vector that can scale the attributes related to that adjective. This vector is then added to the embedding vector of the target subject. The authors also used synonymous versions of the adjective to obtain a more robust vector or optimizing to derive this vector.

**Strengths:**

* The paper is well-written.

* The proposed method is straightforward and intuitive.

* The paper provides a good summary of existing methods and clearly articulates the motivation behind their work.

**Weaknesses:**

* The novelty of the method is limited, and it lacks technical depth. I have some questions regarding the proposed method. First, it seems that the method may not be effective for all prompts. Consider the prompt "two people sitting on the lawn." How should one edit the attributes of just one person? Second, the method heavily relies on the selection of appropriate attribute adjectives, such as "old" and "young" for age. I am curious whether there are some attributes that are difficult to articulate. Intuitively, there are certain characteristics in nature that are currently challenging to describe accurately using human language.

* The experimental results do not support the authors' claims. In Table 1, the method' performance on the subject-specificity metric is weaker than that of the state-of-the-art method, e.g., AdapEdit, and even less effective than simply adding an adjective. Furthermore, when the authors' method is combined with other methods, it appears to degrade their performance. Given these results, I find it difficult to be convinced that the authors' method achieves a competitive performance. Additionally, I would like to know how the reported times in Table 1 are evaluated — do they include the time for training $\Delta e_{A_i}$? Based on my experience, I suspect they do not, which makes the cost comparison unfair. There is also no information on how many instances the metrics are averaged over. By the way, there are too few visual comparisons of different methods, and the authors did not provide results for editing multiple attributes.

**Questions:**

See weaknesses.

---

### Official Review · Reviewer_72Ry · 2024-11-04

**Soundness:** 2
**Presentation:** 3
**Contribution:** 2
**Rating:** 3
**Confidence:** 4

**Summary:**

This work advances the understanding of token-level directions within the commonly used CLIP text embeddings, demonstrating their efficacy for fine-grained, subject-specific control in text-to-image (T2I) models. The authors observed that diffusion models provide a smooth interpretation of CLIP text embedding vectors, allowing for the control of target attributes in generated images by identifying token-level semantic vectors within the CLIP space. Two primary methods are proposed. First, an optimization-free approach is introduced that contrasts embeddings from prompts that describe specific attributes (e.g., "a man" vs. "an old man") to determine the direction vector, with more accurate vectors obtained by averaging the differences across multiple prompt pairs. Second, a learning-based method is employed to identify attribute directions in the noise prediction space of the diffusion model. In summary, this work elucidates the effective application of token-level directions within CLIP embeddings, thereby enabling precise control over multiple attributes of individual subjects in image generation.

**Strengths:**

1. The author identified a token-level attribute vector using a simple yet effective method, which allows control over the representation of specific attributes in the target generated by a diffusion model.
2. Unlike other methods, this approach achieves fine-grained control over specific attributes of the target in the generated images by adding the vector to a specific word in the prompt.
3. Experimental results demonstrate the effectiveness of the proposed method and illustrate its transferability.

**Weaknesses:**

1. The author derives attribute vectors through contrasting descriptions (e.g., "a man" vs. "an old man"). A more intuitive method would involve directly adding the embedding of "old" to "man," yet the author does not compare this simpler approach. It is advisable for the authors to compare their proposed method with this more intuitive approach.
2. While the authors mentioned that direct subtraction of CLIP vectors is only applicable to prefix attributes, the paper does not provide a clear explanation of how the optimization-based method addresses this limitation. Furthermore, the experimental results never examine related content, such as suffix attributes.
3. Will directly adding a vector to the corresponding word completely avoid affecting other targets? During image generation in diffusion models, text tokens interact with pixels through cross-attention layers, and in this process, an attention map is generated, which can reflect the degree to which image pixels focus on a particular word. It is possible that a pixel might exhibit high attention to multiple tokens, which may lead to overlapping influences on the image regions associated with different words. The authors are encouraged to consider such scenario and discuss it.
4. The paper mentions that experiments were conducted using SDXL, which has two text encoders; however, the author does not clearly specify how the attribute vectors are incorporated. Please provide a specific description of how the attribute vectors is incorporated in SDXL's dual text encoders. Additionally, since optimization involved training with images generated by the diffusion model, the model's performance could significantly affect the results. It is recommended that the authors investigate the performance vary across different diffusion models.
5. Based on the results presented in Table 1, the proposed method does not demonstrate exceptional performance across various metrics. Moreover, integrating other methods even leads to a decrease in performance on the subject-specificity metric. The authors are encouraged to discuss the potential reasons for such decreases in performance.
6. The results presented in the paper are based on a few simple attributes and prompts. To enhance the persuasiveness of the method, more complex scenarios should be demonstrated. For example, attributes could involve verbs like " jump" or cases where the target includes multiple words, such as "a red sports car". Additionally, visual comparisons with the baselines should be provided to offer clearer insights into the method’s effectiveness relative to other approaches.

**Questions:**

Please refer to the weakness part.

---

### Official Review · Reviewer_uiRV · 2024-11-04

**Soundness:** 3
**Presentation:** 2
**Contribution:** 2
**Rating:** 5
**Confidence:** 3

**Summary:**

The paper presents an approach to attribute control in text-to-image (T2I) models, addressing a gap in subject-specific and continuous attribute modulation. This work identifies token-level "semantic directions" in the CLIP embedding space, allowing continuous, subject-specific adjustments. The authors propose two techniques to discover these directions: an optimization-free approach using contrastive prompts and a learning-based method employing backpropagation for robust direction identification. The solution integrates seamlessly with existing diffusion models, allowing intuitive and precise control over multiple attributes without modifying the T2I models.

**Strengths:**

1. The concept of leveraging token-level CLIP embeddings for fine-grained, subject-specific control is novel and enhances the flexibility of T2I models in generating nuanced images.
2. The paper provides a comprehensive methodology, with two approaches for direction identification that cater to different application needs. The learning-based approach, particularly, shows robustness and reduces unintended changes.
3. This approach complements current models without the need for model modifications, making it accessible and practical for integration.

**Weaknesses:**

1. This approach is mostly heuristic, and the underlying reasons behind its effectiveness remain unclear.
2. The reliance on linear modulations for semantic directions may limit the flexibility of control in some cases.
3. The robust learning-based method requires backpropagation and multiple training instances, which may be computationally intensive, especially for real-time applications.

**Questions:**

1. The contrastive learning w.r.t. a subject with a prepended adjective $P_+$ is essential to the method. Do you need to manually select the generated images? For instance, do you need to verify if the image generated for "an expensive car" truly appears expensive, or do you let Stable Diffusion determine it automatically? Manually verifying each generated image related to $P_+$ could be highly costly.

2. In Eq.(3), $\lambda$ controls the magnitude of the modulation. Is it possible for $\lambda$ to exceed 1, thereby extending beyond the concept defined by the positive prompt? What is the range of $\lambda$? Any breaking point if $\lambda$ is too large?

3. In Fig.6, what would the distributions look like for negative $\lambda$?

4. The paper suggests that attributes can be adjusted linearly, and that by learning only the positive prompt, the model can naturally generalize to the negative prompt. How can this be explained?

5. How does the model perform in handling multiple complex attributes simultaneously, particularly in scenarios where attribute overlap is subtle?

6. Are there specific attributes or subject types where this approach performs less effectively, and how might these limitations be addressed?

---

### Note · Authors · 2024-11-15

I have read and agree with the venue's withdrawal policy on behalf of myself and my co-authors.